



# Gravitational separation of Ar/N₂ and age of air in the lowermost stratosphere in airborne observations and a chemical transport model

Benjamin Birner[1], Martyn P. Chipperfield[2,3], Eric J. Morgan[1], Britton B. Stephens[4], Marianna Linz[5], Wuhu Feng[2,6], Chris Wilson[2,3], Jonathan D. Bent[1,7], Steven C. Wofsy[5], Jeffrey Severinghaus[1], Ralph F. Keeling[1]

[1]Scripps Institution of Oceanography, UC San Diego, La Jolla, CA 92093, USA
[2]School of Earth and Environment, University of Leeds, Leeds, LS2 9JT, UK
[3]National Centre for Earth Observation, University of Leeds, Leeds, LS2 9JT, UK
[4]National Center for Atmospheric Research, Boulder, CO 80301, USA
[5]Department of Earth and Planetary Sciences, and School of Engineering and Applied Sciences, Harvard University, Cambridge, MA 02138, USA
[6]National Centre for Atmospheric Science, University of Leeds, Leeds, LS2 9JT, UK
[7]now at: Picarro, Inc., Santa Clara, CA 95054, USA

*Correspondence to*: Benjamin Birner (bbirner@ucsd.edu)

**Abstract.** Accurate simulation of atmospheric circulation, particularly in the lower stratosphere, is challenging due to unresolved wave-mean flow interactions and limited high-resolution observations for validation. Gravity-induced pressure gradients lead to a small but measurable separation of heavy and light gases by molecular diffusion in the stratosphere. Because the relative abundance of Ar to N₂ is exclusively controlled by physical transport, the argon-to-nitrogen ratio (Ar/N₂) provides

an additional constraint on circulation and the age of air (AoA), i.e. the time elapsed since entry of an air parcel into the stratosphere. Here we use airborne measurements of N₂O and Ar/N₂ from nine campaigns with global coverage spanning 2008–2018 to calculate AoA and to quantify gravitational separation in the lowermost stratosphere. To this end, we develop a new N₂O-AoA relationship using a Markov Chain Monte Carlo algorithm. We observe that gravitational separation increases systematically with increasing AoA for samples with AoA between 0 to 3 years. These observations are compared to a

simulation of the TOMCAT/SLIMCAT 3-D chemical transport model, which has been updated to include gravitational fractionation of gases. We demonstrate that although AoA at old ages is slightly underestimated in the model, the relationship between Ar/N₂ and AoA is robust and agrees with the observations. This highlights the potential of Ar/N₂ to become a new AoA tracer that is subject only to physical transport phenomena and can supplement the suite of available AoA indicators.

## 1. Introduction

Transport in the middle atmosphere is driven by a combination of advection by the Brewer-Dobson circulation (BDC) (Brewer, 1949; Dobson, 1956) and quasi-horizontal, two-way mixing by breaking waves (Holton et al., 1995). Models consistently



predict an acceleration of the BDC due to climate change (Butchart, 2014) but subgrid-scale mixing processes and momentum transfer by unresolved buoyancy waves limit our ability to accurately simulate circulation in the stratosphere (Haynes, 2005; Plumb, 2007). An acceleration of the BDC has important repercussions for stratosphere-troposphere exchange (STE), and thus

recovery of the ozone layer and the greenhouse effect of stratospheric water vapor, observational evidence for an acceleration of the BDC is weak (Engel et al., 2009, 2017; Waugh, 2009; Ray et al., 2010, 2014).

The mean age of air (AoA) is a widely used indicator of stratospheric circulation (Hall and Plumb, 1994; Waugh and Hall, 2002; Linz et al., 2016). Air can be transported to any location $r$ in the stratosphere via a myriad of different paths, and each path will have an associated transit time. The probability density function that describes the likelihood for air to reach location

$r$ with a specific transit time is called the age spectrum. Although the age spectrum is not directly observable, some aspects of its shape can be inferred from observations of long-lived tracers. For tracers that are conserved in the stratosphere and whose concentrations increase approximately linearly with time in the troposphere, such as $SF_6$ and $CO_2$, the mean AoA, i.e., the first moment of the distribution, can simply be calculated as the time difference, or "lag time", to when tracer concentrations in the upper troposphere last had comparable values as measured in a stratospheric sample (Hall and Plumb, 1994; Boering et al.,

1996; Waugh and Hall, 2002). The stratospheric concentration of $N_2O$ has also been calibrated as an independent AoA tracer by relating the gradual photolytic loss of $N_2O$ in the stratosphere to AoA as determined from $CO_2$ (Boering et al., 1996; Andrews et al., 2001; Linz et al., 2017).

In contrast to early measurements made on rocket samples (Bieri and Koide, 1970), Ishidoya et al. (2008, 2013) have shown using a balloon-borne sampling system that stratospheric air is detectably fractionated by gravitational settling (GS), with the

degree of fractionation strongly correlated to AoA (Ishidoya et al., 2008, 2013, 2018; Sugawara et al., 2018; Belikov et al., 2019). GS leads to depletion of heavier gases in the stratosphere yielding lower ratios of heavy to light gases, e.g. $Ar/N_2$, $^{18}O/^{16}O$ of $O_2$ and $^{15}N/^{14}N$ of $N_2$, with increasing elevation. The vertical gradients induced by gravimetric separation are steeper at higher latitudes (Ishidoya et al., 2008, 2013, 2018; Sugawara et al., 2018), and consistent with patterns observed in stratospheric AoA (Sugawara et al., 2018; Belikov et al., 2019). Gravimetric settling in the stratosphere has also been simulated

in 1-D (Keeling, 1988; Ishidoya et al., 2008), 2-D (Ishidoya et al., 2013, 2018; Sugawara et al., 2018), and 3-D stratospheric models (Belikov et al., 2019). 2-D and 3-D models both show a pattern of GS which increases with altitude and latitude, similar to the patterns observed in tracers with a significant stratospheric sink such as $N_2O$, and consistent with a positive correlation with AoA (Ishidoya et al., 2013; Sugawara et al., 2018; Belikov et al., 2019). Here we attempt to calibrate GS of $Ar/N_2$ as an AoA tracer similar to previous work on the $N_2O$-AoA relationship.

Observing GS in the stratosphere is challenging, however, as the signals are small and because of the need to avoid artifacts caused by temperature- and pressure-induced fractionation near the sampling inlet (Blaine et al., 2006; Ishidoya et al., 2008, 2013). The resulting scatter in  existing balloon-based measurements precludes a clear evaluation of the relationship between GS and AoA (Belikov et al., 2019).





Here we present a dataset of gravitational fractionation of Ar/N$_2$ and AoA observations made on flask samples from three airborne research projects, from 9 campaigns in the lowermost stratosphere with AoA <3 years (Wofsy et al., 2012, 2018; Stephens, 2017). The HIAPER Pole-to-Pole Observations (HIPPO) project was a global survey of the Pacific troposphere to lower stratosphere on the NSF/NCAR Gulfstream-V aircraft (Wofsy et al., 2011), composed of 5 individual campaigns from 2008–2011. The O$_2$/N$_2$ Ratio and CO$_2$ Airborne Southern Ocean (ORCAS) study was conducted using the same aircraft but focused on the Drake Passage and Antarctic Peninsula during Jan–Feb 2016 (Stephens et al., 2018). The Atmospheric Tomography (ATom) project was a survey of the troposphere and lower stratosphere of the Pacific and Atlantic Ocean basins on the NASA DC-8 aircraft, composed of 4 individual campaigns from 2016–2018 (Prather et al., 2017). The observations are compared to new simulations of GS with the TOMCAT/SLIMCAT (Chipperfield, 2006) tracer transport model. Our goals are to demonstrate the consistency of our data with gravitational fractionation, to evaluate model performance, and to highlight the potential of Ar/N$_2$ as a new age tracer.

## 2. Methods

### 2.1. Measurements

Discrete 1.5L flask samples were taken with the NCAR/Scripps Medusa Whole Air Sampler (Bent, 2014) (https://www.eol.ucar.edu/instruments/ncar-scripps-airborne-flask-sampler). Medusa holds 32 borosilicate glass flasks sealed with Viton o-rings and uses active pressure control to fill the flasks with cryogenically dried air to ~760 torr. Flasks are shipped to Scripps Institution of Oceanography (SIO) for analysis of Ar/N$_2$ ratios on an IsoPrime Mass Spectrometer. We report changes in Ar/N$_2$ ratios in delta notation

$$\delta = \left(\frac{R_{SA}}{R_{REF}} - 1\right) \times 10^6 (per\ meg) \qquad\qquad 1$$

where $R_{SA}$ is the mixing or isotope ratio in the sample and $R_{REF}$ the ratio in a reference mixture. Replicate agreement shows a 1σ repeatability of ±6.1 per meg for δ(Ar/N$_2$). For further details, see Bent (2014). Measurements are made on the Scripps O2 Program Argon Scale, as defined on 21 Jan 2020. δ(Ar/N$_2$) values are reported after applying an offset to the data to yield a mean of zero in the equatorial airborne observations of the free troposphere between 2-10 km.

We have compiled available simultaneous, high frequency measurements of a range of other trace gases including N$_2$O, CO$_2$, O$_3$, CH$_4$, and CO to identify Medusa samples with stratospheric influence and calculate AoA. N$_2$O was measured continuously with a precision of 0.09 ppb at 1 Hz frequency using the Harvard Quantum Cascade Laser Spectrometer (QCLS) (Santoni et al., 2014) during HIPPO and ORCAS, and measured every 1–3 min using the NOAA gas chromatograph PAN and Trace Hydrohalocarbon ExpeRiment (PANTHER) during ATom. The Unmanned Aircraft Systems Chromatograph for Atmospheric Trace Species (UCATS) was used to measure O$_3$ during HIPPO and ATom. O$_3$ was not measured on ORCAS. We use



continuous $H_2O$ data from the NCAR open path near-infrared multi-pass spectrometer Vertical Cavity Surface Emitting Laser (VCSEL) (Zondlo et al., 2010) for HIPPO and ORCAS whereas $H_2O$ was measured using the NASA Diode Laser Hygrometer (DLH) (Diskin et al., 2002) during ATom flights. $CH_4$ and CO were measured by QCLS during HIPPO and ORCAS, and by

the NOAA Picarro (Karion et al., 2013) during ATom. An averaging kernel is applied to the continuous and semi-continuous aircraft data, such as $N_2O$, $O_3$, and $H_2O$, to match it to Medusa samples. The kernel multiplies a weighting function $w_i(t)$ with all continuous data before time $t_s$ when Medusa switched from sample flask $i$ to the next. $w_i(t)$ for each sample $i$ is given by

$$w_i(t) = \exp\left[\frac{-t_s - t}{\tau}\right], \qquad\qquad 2$$

where $t$ is each 1s-increment of the continuous data and $\tau = \frac{V}{Q}$ is the flushing time of air in a Medusa flask determined by the flask volume $V$ and airflow $Q$.

Stratospheric samples are identified based on their $N_2O$, $O_3$, and water vapor levels. Classification based on chemical composition rather than potential temperature or altitude effectively selects samples with a clear stratospheric signature in the lowermost stratosphere, where mixing with the upper troposphere could be substantial, and minimizes the impact of synoptic-scale variability. We label samples as "stratospheric" if (i) water vapor levels are below 50 ppm and either (ii.a) $O_3$ values exceed 140 ppb or (ii.b) $N_2O$ (detrended to a reference year of 2009) is below 315 ppb. These criteria yield 235 lower

stratospheric samples with high quality $N_2O$ and $\delta(Ar/N_2)$ data, spanning a wide range of latitudes poleward of 40° in both the Northern and Southern Hemisphere (Fig. 1). We use Medusa samples from all 5 HIPPO campaigns, ORCAS, and ATom 2–4. We do not use samples from ATom 1 because inlet fractionation as a combined result of the unique inlet design and location on the DC-8 on this campaign introduced apparent biases on the order of 30 per meg. An additional 14 stratospheric samples are available from the START-08 campaign on the NSF/NCAR GV, but we have not used these here because the $\delta(Ar/N_2)$

data quality is considerably worse.

### 2.2. AoA calculation

Stratospheric AoA is calculated from $N_2O$ using an updated hemisphere-specific $N_2O$-AoA relationship. Our method broadly follows Andrews et al. (2001) who assumed a bimodal age spectrum and use multiple observations of $CO_2$ binned by $N_2O$ values to resolve the seasonal cycle of $CO_2$ in each bin. Properties of the age spectrum for each $N_2O$ bin, including AoA were

constrained by optimizing the agreement between observed $CO_2$ concentrations and concentration implied by randomly generated age spectra in each $N_2O$ bin. Andrews et al. (2001) used a highly efficient "genetic algorithm" to yield the most likely relationship between AoA and $N_2O$ in each bin. In contrast, we use a more computationally costly Markov chain Monte Carlo (MCMC) method that allows us to obtain more robust uncertainties for all estimated parameters of the age spectrum. Following Malinverno (2002) and Green (1995), our algorithm builds on a Metropolis-Hasting sampler (Metropolis et al.,

1953; Hastings, 1970) to evaluate probability distributions for each age spectrum parameter and automatically choses whether



a unimodal or bimodal representation of the age spectrum is more appropriate in each $N_2O$ bin. Finally, we constructed a new tropical upper troposphere reference time series for $CO_2$ in this study to ensure maximum consistency between all observations used. Analytical and methodological uncertainties are propagated thoroughly and reported as the 95% confidence interval around a mean.

Our upper troposphere reference time series (TRTS) consists of a long-term trend and a representation of the mean seasonal cycle. Because direct $CO_2$ observations in the region are limited, the long-term trend and a first guess of the mean seasonal cycle are estimated from monthly mean surface observations at the Mauna Loa station in Hawaii (MLO, 19°N, 155°W, Fig 2, panel (a)) (Keeling et al., 2001) which are later adjusted to match airborne observations in the tropics (20°S-20°N) above 8 km (Fig. 2, panel (c)). The adjustment includes (i) a constant offset, (ii) a reduced amplitude of the seasonal cycle and (iii) a

phase-lag of one month. As shown in Fig. 2, after these adjustments the TRTS matches the mean seasonal cycle and absolute value of the airborne data, thus accounting for known vertical gradients of $CO_2$ and reduced seasonality in the upper troposphere (Fig. 2, panel (d)). The amplitude of the seasonal cycle in our time series is also in good agreement with the boundary condition used by Andrews et al. (1999 and 2001).

     The new TRTS is used to estimate the age spectrum in 13 $N_2O$ bins of 5 or 10 ppb width (320-325 ppb,…, 290-295 ppb, 280-

290,…, 230-240 ppb) using a Markov Chain Monte Carlo (MCMC) algorithm which compares observed $CO_2$ concentrations in the bin to concentration expected from the TRTS. To maximize data availability, we use high frequency data (see "2.1. Measurements") identified as stratospheric according to the criteria above from the 10 s merged products available for HIPPO, ORCAS, and ATom rather than data averaged to the lower Medusa sampling interval for this calibration exercise. Small corrections are applied to the observed $CO_2$ concentrations (<0.2 ppm) to account for oxidation of $CH_4$ and CO in the

stratosphere.

     The MCMC algorithm considers random noise in the TRTS and uses a unimodal or bimodal inverse Gaussian shape of the age spectrum characterized by mean ages $\Gamma_1$ and $\Gamma_2$ and shape parameters $\lambda_1$ and $\lambda_2$

$$G\left(t'|(\Gamma, \lambda)\right) = A \sqrt{\frac{\lambda_1}{2\pi t'^3}} \exp\left(-\frac{\lambda_1 \, (t' - \Gamma_1)^2}{2\Gamma_1^2 t'}\right) + (1 - A) \sqrt{\frac{\lambda_2}{2\pi t'^3}} \exp\left(-\frac{\lambda \, (t' - \Gamma_2)^2}{2\Gamma_2^2 t'}\right), \qquad\qquad 3$$

where factor $A$ determines the relative weight of each peak and a value of $A = 1$ yields a unimodal spectrum (Hall and Plumb, 1994; Andrews et al., 2001). By setting $A = 1$ for 50% of all tested age spectra, the MCMC algorithm automatically selects

whether a unimodal or bimodal representation of the age spectrum is optimal to match observations. Although bimodal solutions with five instead of two free parameters will always be able to fit the data better, a larger number of parameters also decreases the likelihood of randomly selecting a combination of parameters that match the observations well because the fraction of the total parameter space region which yields good agreement with the observation decreases as more parameters are added (Malinverno, 2002). The algorithm is thus able to make an appropriate choice between unimodal and bimodal





distributions from the data itself, without any further a priori assumptions. The MCMC algorithm is set up separately with 2000 independent chains for each $N_2O$ bin to account for uncertainty in the TRTS and obtain best estimates of the mean AoA, i.e., the first moment of Eq. 3. To simplify the algorithm, possible values of $\Gamma_1, \Gamma_1, \lambda_1$ and $\lambda_2$ are repeatedly sampled from the same $N_2O$ bin-specific prior distributions. Details of the algorithm are presented in Appendix A.

Finally, the resulting relationship between mean AoA and the $N_2O$ concentration of each bin is fit separately for each
hemisphere by a quadratic polynomial and the polynomial is evaluated at the $N_2O$ value of each Medusa sample to pair every observation of $\delta(Ar/N_2)$ with an AoA. Uncertainty in $N_2O$ and the polynomial fits are propagated by a Monte Carlo scheme. Overall, our method estimates the most likely mean AoA for each Medusa sample and improves upon previous methods by providing a thorough treatment of uncertainty resulting from (i) analytical error, (ii) uncertainty in the shape of the age spectrum, and (iii) uncertainty in the composition of source gas introduced into the stratosphere

**2.3. TOMCAT/SLIMCAT model**

TOMCAT/SLIMCAT (hereafter TOMCAT) is an offline 3-D chemical tracer transport model that has been used extensively for studies of stratospheric ozone depletion and circulation (e.g., Chipperfield, 2006; Chipperfield et al., 2017; Krol et al., 2018). For this study, TOMCAT was run over 31 years, from 1988 to 2018, with a timestep of 30 minutes at $2.8° \times 2.8°$ horizontal resolution forced by the ERA-Interim reanalysis (Dee et al., 2011) at 60 vertical hybrid sigma- pressure ($\sigma$–p) levels
up to ~60–65 km. The first 20 years (i.e. before the first flask observation) were treated as spin-up. The TOMCAT AoA tracer is initialized at the surface and corrected to a value of AoA = 0 just below the tropical tropopause. Vertical motion was calculated from the divergence of the horizontal mass fluxes. Although this approach gives slightly younger stratospheric AoA than using isentropic levels and radiative heating rates, it allows a more detailed treatment of tropospheric transport (Chipperfield, 2006; Monge-Sanz et al., 2007). The model simulation was sampled at the times and locations of the Medusa
flask observations to provide a direct comparison between the measurements and model.

Following the methodology of Belikov et al. (2019), we include an additional vertical flux term in the model representing the GS of gases in the atmosphere. The vertical flux $f_i$ (mol m$^{-2}$ s$^{-1}$) of tracer $i$ due to molecular diffusion in Earth's gravitational field is given as (Banks and Kockarts, 1973; Belikov et al., 2019)

$$f_i = -D_i N C_i \left[ \frac{1}{C_i} \frac{\partial C_i}{\partial z} + \left( \frac{1}{H_i} - \frac{1}{H_{air}} \right) + \alpha_i \frac{1}{T} \frac{\partial T}{\partial z} \right], \qquad 4$$

where $D_i$ is the tracer-specific binary molecular diffusivity in air (m$^2$ s$^{-1}$), $N$ is the number density of air (mol m$^{-3}$), $C_i$ is the
mixing ratio, $H_i = \frac{RT}{gM_i}$ is the tracer-specific atmospheric equilibrium scale height (m), $R$ is the fundamental gas constant (J K$^{-1}$ mol$^{-1}$), $g$ is the gravitational constant (m s$^{-2}$), $M_i$ is the tracer-specific atomic or molecular mass (kg mol$^{-1}$), $\alpha_i$ is the tracer-specific thermal diffusivity (m$^2$ s$^{-1}$), and $T$ is temperature (K). The three terms in Eq. (4) represent molecular diffusion driven by (i) vertical gradients in the mole fraction of $i$, (ii) pressure gradients caused by gravity and described by the barometric law,



and (ii) temperature gradients (left to right). We neglect the first and third terms in the brackets in Eq. (4), leaving only the

gravitational settling term $\left(\frac{1}{H_i} - \frac{1}{H_{air}}\right)$ on the basis that both terms are orders of magnitude less important than the gravitational

separation term under stratospheric conditions (see Appendix B) (Ishidoya et al., 2013; Belikov et al., 2019). No fluxes are

allowed through the top boundary, and $C_i$ is held constant at the Earth's surface.

To simplify the numerical treatment, we only simulate a single tracer of gravitational fractionation in the atmosphere $\delta_{GST}$

with a molecular mass 1 amu greater than that of air which can be scaled offline (see Appendix B & C) to obtain $\delta(Ar/N_2)$

using

$$\delta(Ar/N_2) \approx \frac{(M_{Ar} - M_{air}) \times D_{Ar} - (M_{N2} - M_{air}) \times D_{N2}}{(M_{GST} - M_{air}) \times D_{GST}} \times \delta_{GST}. \qquad 5$$

The appropriate diffusivity values $D_{Ar}$ and $D_{N2}$ for Ar and $N_2$ in air are derived in Appendix B for a ternary mixture of Ar, $O_2$,

and $N_2$, extending previous work (Ishidoya et al., 2013; Belikov et al., 2019).

### 2.4. NIES TM model

We compare our results to previous simulations of GS using the National Institute for Environmental Studies chemical

transport model (NIES TM) published recently by Belikov et al. (2019). The NIES TM is a three-dimensional transport model

of similar complexity as TOMCAT driven by the Japanese 25-year Reanalysis (JRA-25) with a hybrid sigma–isentropic (σ–

θ) vertical coordinate up to 5 hPa or ~35 km. The model and GS results are described in detail in Belikov et al. (2013) and

Belikov et al. (2019), respectively.

### 3.   Results

### 3.1.   $N_2$O-AoA calibration

Our new $N_2$O-AoA relationships for the Northern and Southern Hemisphere (NH & SH) are well constrained by the

observations and generally follow the mid-latitude NH calibration curve of Andrews et al. (2001) (Fig. 3, panel (d)). At

AoA>2.5, the $N_2$O-AoA relationships yields slightly younger ages in the NH than in the SH, and compared to the previously

published curve, suggesting there might be a latitudinal dependence of the relationship. Such a latitude-dependence is, in fact,

expected based on theory (Plumb, 2007). We expect curvature and a latitudinal difference in the $N_2$O-AoA relationship because

photolysis of $N_2$O depends on latitude and altitude due to local sunlight availability. Furthermore, mixing of young and old air





results in a mixture with anomalously low $N_2O$ concentration for a given age. Since the SH, NH and tropics feature different photolysis rates and show different degrees of mixing/isolation, different $N_2O$-AoA relationship are expected.

Unimodal age spectra are preferentially selected for young AoA (Fig. 3, panels (a)-(c)) whereas bimodal spectra are slightly
more common for older samples representing 50-80% of the solution ensemble in these bins. However, confidence intervals on age spectra parameters from bimodal spectra are considerably wider than for unimodal spectra. This implies that the parameters in a bimodal distribution are redundant and not sufficiently constrained by the observations used. It appears that not enough data is available from the airborne campaigns to determine the amplitude of the seasonal cycle with enough confidence to distinguish the relative contribution of the old peak (influencing only the mean concentration difference to the
troposphere) and the young peak (controlling the amplitude of seasonality) in bimodal age spectra. Because bimodal distributions are generated with a random value of the weighting factor $A$ and the mean AoA of the second peak is assumed to be old (5-7 years), randomly generated bimodal solutions often produce overall AoA that is quite old. Therefore, they are less likely to be selected by the MCMC algorithm for young AoA bins and a prevalent occurrence of unimodal age spectra in these bins is expected by chance. If more stratospheric data were available, the seasonality of $CO_2$ in each $N_2O$ bin would be better
resolved and the algorithm could derive tighter constraints on all parameters in bimodal age spectra, also allowing it to distinguish more clearly between unimodal or bimodal age spectra. Despite the limited resolution of the seasonal cycle, the observations are sufficient to place tight limits on the AoA in each $N_2O$ bin and yield a well-characterised relationship between $N_2O$ and AoA for each hemisphere.

### 3.2.   The relationship between AoA and GS in models and observations

A comparison of the AoA-GS relationship with observations yields good agreement for the TOMCAT model results within the observational uncertainties (Fig. 4 panel (a)), but the observations fall outside the range of GS predicted by the NIES TM (Belikov et al., 2019) for AoA >1 year (Fig. 4, panel (b)). For young samples with AoA < 3 years, GS of $\delta(Ar/N_2)$ increases by roughly 35-45 per meg per year of AoA in both TOMCAT and observations, and converges to zero for the youngest samples. In the upper stratosphere, TOMCAT does not obtain any ages as old as observed by Ishidoya et al., (2008, 2013,
2018) and therefore cannot reproduce these observations directly. Changing the vertical coordinate system of TOMCAT or forcing the model with a different reanalysis product could improve agreement with the observations for old ages because TOMCAT in the configuration used here is known to slightly underestimate AoA in the upper stratosphere (Monge-Sanz et al., 2007; Chabrillat et al., 2018). The very steep AoA-GS relationship for the oldest simulated air is however seen in TOMCAT and the balloon observations. Overall, our observations validate the implementation of GS for young (< 3 y) ages in TOMCAT.

The relationship between AoA and GS differs between TOMCAT and the NIES TM (Fig. 5). The NIES TM shows weaker curvature in the relationship overall and produces larger declines in $Ar/N_2$ at ages less than 4.5 years compared to TOMCAT. In TOMCAT's mesosphere (which is at the limit of the domain covered by the ERA-Interim reanalyses forcing the model), AoA is near uniform but GS continues to increase with increasing altitude, changing the relationship between AoA and GS in

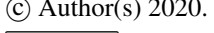



this region. The AoA-GS relationship in TOMCAT is very similar in all non-tropical regions (outside 15° > lat > -15°) whereas

the curvature of the relationship is slightly stronger in the tropics. In contrast, the NIES TM has a clear dependence of the AoA-GS relationship on latitude. There is also some evidence in the observations for a dependence of the AoA-GS relationship on hemisphere. In the observations, $\delta(Ar/N_2)$ values appear to be slightly more negative in the SH than in the NH for the same age, particularly for AoA > 1.5 years. However, almost all SH samples with AoA older than 1.5 years come only from the ORCAS campaign and the scatter in our observations generally makes it difficult to separate signal from noise for small

interhemispheric differences.

### 3.3.  GS and AoA in TOMCAT

Annual mean $\delta(Ar/N_2)$ in TOMCAT follows the typical pattern of a tracer with a stratospheric sink (Chipperfield, 2006), as previously found in simulations using the NIES TM (Belikov et al., 2019) and the SOCRATES model (Ishidoya et al., 2013; Sugawara et al., 2018) (Fig. 6). $\delta(Ar/N_2)$ is zero in the troposphere and decreases with elevation. The most depleted $\delta(Ar/N_2)$

is observed at high latitudes where sinking air of the Brewer-Dobson circulation advects strongly fractionated air downward. In the tropics, $\delta(Ar/N_2)$ values are considerably less fractionated at the same altitude due to upwelling of unfractionated tropospheric air. Vertical gradients in $\delta(Ar/N_2)$ generally increase with altitude and are largest in the mesosphere (> 50km) because molecular diffusion increases with decreasing pressure and eventually dominates above the turbopause (not shown). There are strong seasonal changes in $\delta(Ar/N_2)$ depletion on the order of several thousand per meg, in particular in the high

latitude mesosphere, with the strongest fractionation occurring during the winter season (see movie available with online version of the manuscript).

The AoA tracer in TOMCAT shows a similar pattern to $\delta(Ar/N_2)$ with younger ages at low altitude and in the tropics and oldest ages in the mesosphere. Vertical gradients in AoA are largest at high latitudes close to the tropopause. At high latitudes above 20 km and at low latitudes above 50 km, vertical gradients of AoA mostly disappear and AoA becomes nearly uniform.

## 4.  Discussion

### 4.1.  Difference between TOMCAT and the NIES TM

We hypothesize that an adequate representation of the mesosphere in models is critical in determining the curvature of the AoA-GS relationship. The residence time of air above ~40 km is rather short in TOMCAT and AoA is nearly constant with altitude in this region. GS in contrast continues to increase with altitude because of the diffusivity dependence on pressure

allowing gases to separate more effectively. The much lower top of the NIES TM (~35 km vs 60–65 km) reduces its ability to capture this effect, which impacts the AoA-GS relationship because the mesospheric signal is exported into the stratosphere, in particular in polar regions where mesospheric air is sinking. TOMCAT furthermore produces a less negative slope of the AoA-GS relationship for young air (AoA <3 years) and greater similarity in the AoA-GS relationship between latitude bands

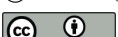



than the NIES TM as shown in Fig. 5. This could in part be a consequence of using a different meteorological reanalysis
product for forcing the two models (Chabrillat et al., 2018), or could indicate differences between the models in vertical and
horizontal mixing intensity.

### 4.2.  Estimating AoA from observed $\delta(Ar/N_2)$

Different tracers of AoA all have unique strengths and weaknesses. Estimating AoA in the lowermost stratosphere from $CO_2$
for example is limited by complexities involving seasonality and the possibility of multiple entry points of air into the
stratosphere due to isentropic mixing with the midlatitude troposphere (Hall and Plumb, 1994; Andrews et al., 2001; Waugh
and Hall, 2002). $SF_6$-derived AoA in contrast is biased high at mid and high latitudes due to the influence of mesospheric air
in which $SF_6$ has been depleted by electron attachment, photolysis, and chemical reactions (Kovács et al., 2017; Linz et al.,
2017). $N_2O$ as an AoA tracer relies on the photolytic destruction of $N_2O$ in the stratosphere which may depend strongly on
location and had only been empirically calibrated for young ages at mid latitudes so far (Andrews et al., 2001; Linz et al.,
275   2017).

Thanks to the robust relationship between AoA and $\delta(Ar/N_2)$, and the small seasonal cycle amplitude of $\delta(Ar/N_2)$ <6 per meg
in the upper troposphere (Fig. S2), AoA could also be estimated from $\delta(Ar/N_2)$. Using the current analytical $\delta(Ar/N_2)$
uncertainty of 12.2 per meg ($2\sigma$) and the AoA-$\delta(Ar/N_2)$ relationship seen in TOMCAT (including variability from seasonal
and latitudinal differences), we estimate that AoA could be calculated to within about $\pm0.4$ years ($2\sigma$, Fig. S1). This is still
considerably worse than the $\pm0.1$ years confidence interval in AoA estimated from $N_2O$. However, the uncertainty is almost
exclusively analytical and can be improved with future improvements in sampling and the accuracy and precision of the
measurements.

The heavy noble gases krypton and xenon will be roughly 3.6× and 5.8× more strongly fractionated in the stratosphere than
Ar, but are also more challenging to measure due to their ~8000× and 100000× lower abundance in the atmosphere.
Nevertheless, future analysis of these gases in stratospheric air samples might further improve our ability to estimate AoA
from the gravitational fractionation of gases and help diagnose artefactual fractionation, because heavier noble gases are more
strongly fractionated under the influence of gravity and less sensitive to thermal fractionation (Seltzer et al., 2019).

### 4.3.  Future directions

An open question in climate applications of noble gases is whether there could be a stratospheric influence on tropospheric
$\delta(Ar/N_2)$. Tropospheric observations of $\delta(Ar/N_2)$ and other noble gas-elemental ratios have been used to infer ocean heat
content changes by capitalizing on the temperature-dependency of gas solubility in the oceans (Keeling et al., 2004; Headly
and Severinghaus, 2007; Ritz et al., 2011; Bereiter et al., 2018). However, long-term trends (Butchart, 2014) and natural
variability in stratospheric circulation and stratosphere-troposphere exchange (STE) such as the Quasi-Biennial Oscillation
(QBO) (Baldwin et al., 2001) could advect a stratospheric GS signal into the troposphere and alias onto surface observations





of these gases. To this end, we calculate the volumetric Ar deficit in the atmosphere in moles using TOMCAT (Fig. 7) relative

to a hypothetical, unfractionated atmosphere with a homogenous mixing ratio ($C_{o_{Ar}}$) at all elevations. Ar deficit is

$$Ar_{deficit} = \left(C_{Ar} - C_{o_{Ar}}\right) \times N_{air} \qquad\qquad 6$$

where $C_{Ar}$ is the simulated Ar mixing ratio and $N_{air}$ is the number density of air.

The Ar deficit is concentrated in the lower stratosphere at around 20 km and at mid to high latitudes. Although the GS signal

is considerably stronger in the mesosphere, the potential for perturbing the troposphere is low given the low molar density of

air in the mesosphere. The region with greatest potential to influence the troposphere therefore lies in the lower stratosphere.

The total Ar deficit of the atmosphere above 200 hPa is approximately -3.9 $\times 10^{13}$ mol and the atmosphere below 200 hPa

contains roughly 1.3 $\times 10^{18}$ mol of Ar. Perturbing STE and/or the stratospheric circulation by 10%, consistent with interannual

to decadal variability of STE in models (Salby and Callaghan, 2006; Ray et al., 2014; Montzka et al., 2018), thus should lead

to a detectable signal of roughly 3 per meg (-3.9 $\times 10^{13}$/1.3$\times 10^{18}\times$10%$\times 10^{6}$) in tropospheric $\delta$(Ar/N$_2$). Corresponding advection

of stratospheric GS signals in N$_2$ amplifies the pure Ar signal by roughly 10%. A careful investigation of such a signal in the

$\delta$(Ar/N$_2$) surface network data (Keeling et al., 2004) is needed because secular trends of $\delta$(Ar/N$_2$) caused by degassing of Ar

and N$_2$ from a warming ocean are also expected to be on the order of 2-3 per meg per decade. Previous studies have also used

ratios involving heavier noble gases (Xe/N$_2$, Kr/N$_2$) to reconstruct mean ocean temperature changes over glacial-interglacial

timescales (Headly and Severinghaus, 2007; Bereiter et al., 2018; Baggenstos et al., 2019). Simultaneous changes in

stratospheric circulation and STE affect heavier noble gases more strongly than Ar/N$_2$ and will need to be accounted for in

such applications of noble gas thermometry.

## 5.  Conclusion

With improvements in data treatment, measurement quality, and modelling constraints, we have shown that gravitational

fractionation of Ar relative to N$_2$ in the stratosphere and mesosphere is a potentially powerful constraint on circulation. High-

precision observations of $\delta$(Ar/N$_2$) in air samples of the lowermost stratosphere from 9 airborne campaigns are well captured

by the 3-D chemical transport model TOMCAT/SLIMCAT, which has been updated to account for the influence of gravity on

air composition through molecular diffusion. In the observations and the model, $\delta$(Ar/N$_2$) is directly related to stratospheric

age of air (AoA) derived here using a new calibration of N$_2$O. Our observations for AoA <3 years produce a slope of roughly

35-45 per meg of $\delta$(Ar/N$_2$) per year of AoA. TOMCAT/SLIMCAT shows better agreement with the new observations than

the NIES transport model (Belikov et al., 2019) and we speculate that the model disagreement could be explained by (i) the

factor of 2 lower top of the NIES transport model, (ii) the use of different reanalysis products, and/or (iii) differences in vertical

and horizontal mixing. In this context, further work is also needed to study the influence of unresolved turbulence on AoA and

$\delta$(Ar/N$_2$) in chemical transport models.





As the importance of stratospheric circulation for ozone recovery, climate projections, and evaluation of tropospheric trends
in halocarbons is increasingly recognized, a need for new observations from the still undersampled stratosphere is becoming
evident. Combining $\delta(Ar/N_2)$ with other tracers of circulation could lead to new insights into atmospheric mixing and transport.
$\delta(Ar/N_2)$ has potential advantages over existing approaches based on transient tracers such as $CO_2$ or $N_2O$ since $\delta(Ar/N_2)$ is
only influenced by the physics of transport and mostly unaffected by seasonality. Furthermore, because of its large gradients
at high altitudes, $\delta(Ar/N_2)$ observations from the upper stratosphere and mesosphere could improve our understanding of
circulation on seasonal and interannual timescales in a region where changes in AoA from transient tracers are otherwise
difficult to resolve.

## 6. Data availability

All Data from the ORCAS, HIPPO and ATom airborne campaigns are freely available at: doi:10.3334/CDIAC/HIPPO_014,
doi:10.5065/D6SB445X, and doi:10.3334/ornldaac/1581. All of the primary data used here is consistent with the associated
merged data products in Wofsy et al. (2012), Stephens (2017), and Wofsy et al. (2018). For Medusa samples identified as
stratospheric, $CO_2$, $\delta(Ar/N_2)$, AoA, and kernel-averaged $N_2O$, $O_3$, $H_2O$, CO, $CH_4$, as well as additional metadata are available
as a supplement to this manuscript at xxxx (doi: in prep.).

NIES TM modelling results (Belikov et al., 2019) and stratospheric observations of GS from the cryogenic balloon sampling-
system (Ishidoya et al., 2008, 2013, 2018; Sugawara et al., 2018) were provided directly by the authors.

## 7. Appendix A: Description of the Markov chain Monte Carlo algorithm

The following list outlines key steps in our MCMC algorithm to calculate AoA from $CO_2$ for each $N_2O$ bin:

1) Start a new Markov Chain and allow for uncertainty in TRTS by adding white noise with an amplitude given by the
scatter of upper tropospheric observations around the mean TRTS in Fig. 2.

2) With a 50% chance, peak weighting factor $A$ is set to zero and a unimodal spectrum is tested (k=1). Alternatively, $A$
is allowed to vary between 0 and 1 for a bimodal distribution (k=2).

3) The other age spectrum parameters are selected from $N_2O$ bin-dependent prior distributions: $\Gamma_1$ is sampled from a
uniform prior distribution with a mean AoA predicted by the $N_2O$-AoA relationship of Andrews et al. (2001) and
generous width (>1.5 years). $\Gamma_2$ is sampled from a uniform prior distribution with values between 5-7 years based on
the nearly invariant value of ~6 years previously found by Andrews et al. (2001). The shape parameters are defined
as $\lambda_i \equiv \frac{\Gamma_i^2}{2\gamma_i}$, and $\gamma_i$ is chosen randomly for each peak with values between 0.1 and 1, as observed in previous studies
(Hall and Plumb, 1994; Andrews et al., 2001; Waugh and Hall, 2002).





4) Convolve the age spectrum calculated from the parameters ($\boldsymbol{m} \equiv [\Gamma_1, \Gamma_2, \lambda_1, \lambda_2, A]$) with the perturbed TRTS to obtain a possible time series of $CO_2$ in the $N_2O$ bin and calculate the misfit ($\boldsymbol{e}$) between the $CO_2$ time series and observations.

5) Calculate the likelihood function $P(\boldsymbol{d}|k, \boldsymbol{m})$ for the set of parameters $\boldsymbol{m}$ and $k$, given a total of $n$ observations ($\boldsymbol{d}$)

$$P(\boldsymbol{d}|k, \boldsymbol{m}) = \frac{1}{[(2\pi)^n \det(\widehat{\boldsymbol{C}}_e)]^{0.5}} \exp\left(-\frac{1}{2} \boldsymbol{e}^T \widehat{\boldsymbol{C}}_e^{-1} \boldsymbol{e}\right) \qquad 7$$

where $\widehat{\boldsymbol{C}}_e$ is the covariance matrix. Because all $n$ observations are independent, $\widehat{\boldsymbol{C}}_e$ has only diagonal entries of $\sigma_{CO_2}^2$ and $\det(\widehat{\boldsymbol{C}}_e)$ simplifies to $(\sigma_{CO_2}^2)^n$. The value of $\sigma_{CO_2}^2$ is different for each bin and determined iteratively as the approximate root mean square error of the observations around the final time series for each bin obtained at the end of the MCMC algorithm. Typical values of $\sigma_{CO_2}^2$ are between 0.18 and 1.28 ppm and generally decrease with increasing AoA of a $N_2O$ bin.

6) If this is the first pass of the chain, define $k_{saved}$ and $\boldsymbol{m}_{saved}$ to equal $k$ and $\boldsymbol{m}$. Otherwise, calculate the selection criterion $\alpha \equiv \min\left(1, \frac{P(\boldsymbol{d}|k, \boldsymbol{m})}{P(\boldsymbol{d}|k_{saved}, \boldsymbol{m}_{saved})}\right)$ and accept $k$ and $\boldsymbol{m}$ as new saved values ($k_{saved}, \boldsymbol{m}_{saved}$) with probability $\alpha$. Sampling from the same prior distributions on each pass of the chain simplifies our expression of $\alpha$ compared to that presented by Malinverno (2002), making it only dependent on the likelihood ratio.

7) Repeat steps 2-6 1000 times sampling parameter values from the same prior distributions and store the final values
of $k_{saved}$ and $\boldsymbol{m}_{saved}$ obtained after the 1000th iteration (i.e., a plausible solution produced past the burn-in period) for later use.

8) To sample the full posterior pdf (i.e., the full uncertainty about the age spectrum parameters), initialize 2000 different Markov chains by repeating steps 1-7. Each stored value of $\boldsymbol{m}$ characterizes one age spectrum that is likely not far from the best solution, given the data $\boldsymbol{d}$, yielding an ensemble of 2000 age spectra from which statistics can be
computed. Note that each Markov chain is fully independent, so the algorithm can be easily parallelized to minimize computational costs.

## 8. Appendix B: Derivation of Eq. (4) from the Maxwell-Stefan Equations

We start by approximating air as a ternary mixture of $N_2$, $O_2$ and Ar, and later generalize to consider additional trace species. According the Maxwell-Stefan equations (Taylor and Krishna, 1993) diffusion in this ternary mixture is governed by:

$$d_{N2} = \frac{C_{N2} f_{Ar} - C_{Ar} f_{N2}}{N \times D_{N2:Ar}} + \frac{C_{N2} f_{O2} - C_{O2} f_{N2}}{N \times D_{N2:O2}} \qquad (A1)$$

$$d_{Ar} = \frac{C_{Ar} f_{N2} - C_{N2} f_{Ar}}{N \times D_{Ar:N2}} + \frac{C_{Ar} f_{O2} - C_{O2} f_{Ar}}{N \times D_{Ar:O2}} \qquad (A2)$$





$$f_{O2} = -f_{N2} - f_{Ar} \tag{A3}$$

$$C_{N2} + C_{O2} + C_{Ar} = 1 \tag{A4}$$

where $C_i \equiv n_i/N$ is the mole fraction, $n_i$ is molar or number density (mol m$^{-3}$), $N$ is the total number density (mol m$^{-3}$), $f_i$ is the molecular diffusion flux (mol m$^2$ s$^{-1}$) relative to the molar average velocity of the mixture, $d_i$ is the thermodynamic driving force for molecular diffusion (m$^{-1}$), and $D_{i:j}$ is the binary diffusion coefficient of the $(i:j)$ pair. An equation for O$_2$ that is analogous to Eq. (A1) is not needed because changes in O$_2$ are governed by the conservation Eqs. (A3) and (A4). Binary diffusivity coefficients (cm$^2$ s$^{-1}$) can be calculated using the method of Fuller et al. as reported in (Reid et al., 1987)

$$D_{ij} = 0.001 \frac{T^{1.75} \left( \frac{1}{M_i} + \frac{1}{M_j} \right)^{0.5}}{P \left[ v_i^{1/3} + v_j^{1/3} \right]^2} \tag{A5}$$

where $P$ is pressure (atm), $v_i$ is the molecular diffusion volume of a trace gas or air (Table A1), and $M_i$ is the molecular mass (g mol$^{-1}$) of a gas.

For an ideal gas, $d_i$ is given by

$$d_i \equiv \nabla C_i + (C_i - \omega_i) \frac{\nabla P}{P} - \frac{\rho_i}{P} \left( \frac{F_i}{M_i} - \sum_{j=1}^{n} \omega_i \frac{F_j}{M_j} \right) + \frac{k_i^T}{T} \nabla T \tag{A6}$$

where $\omega_i \equiv \frac{\rho_i}{\rho} = \frac{M_i C_i}{M_{air}}$ is the mass fraction of gas $i$, $\rho_i$ is density of $i$ (kg m$^{-3}$), $P$ is pressure (Pa), $T$ is temperature (K), $k_i^T$ is the thermal-diffusion ratio of $i$, $M_i$ is the molecular mass of $i$ (kg mol$^{-1}$), and $F_i$ the external body force per mole (N mol$^{-1}$) for
$i$ (Chapman et al., 1990; Taylor and Krishna, 1993).

In the atmosphere, (vertical) pressure gradients are caused by gravity and well approximated by hydrostatic balance

$$\nabla P \approx \frac{\partial P}{\partial z} \approx -\rho g = -\frac{P M_{air}}{RT} g \tag{A7}$$

The gravitational force per mole $F_i$ is

$$F_i = -\frac{\rho_i}{n_i} g \tag{A8}$$

Substituting Eqs. (A7) and (A8) into Eq. (A6) yields





$$d_i \approx \nabla C_i - \left(1 - \frac{M_i}{M_{air}}\right) C_i \frac{1}{P} \frac{PM_{air}g}{RT} + \frac{\rho_i}{P}\left(\frac{\frac{\rho_i}{n_i}g}{M_i} - \sum_{j=1}^{n} \omega_j \frac{\frac{\rho_j}{n_j}g}{M_j}\right) + \frac{k_i^T}{T}\nabla T$$

$$= \nabla C_i + (M_i - M_{air})\frac{g}{RT}C_i + \frac{\rho_i}{P}\left(g - \sum_{j=1}^{n}\omega_i g\right) + \frac{k_i^T}{T}\nabla T \qquad (A9)$$

$$= \nabla C_i + \left(\frac{1}{H_i} - \frac{1}{H_{air}}\right)C_i + \frac{k_i^T}{T}\nabla T$$

where we use the definition of the scale height $H_i \equiv \frac{RT}{gM_i}$. The two terms involving the body force cancel because all species
experience the same gravitational force per unit mass. The tendency for gravimetric separation instead arises from the pressure
gradient term proportional to $\left(\frac{1}{H_i} - \frac{1}{H_{air}}\right)$.

Equations (A1) and (A2) can be inverted to solve for $f_{N2}$ and $f_{Ar}$ (Taylor and Krishna, 1993)

$$f_{N2} = -ND_{N2}^{air}d_{N2} - ND_{N2\times(Ar,O2)}^{air}d_{Ar} \qquad (A10)$$

$$f_{Ar} = -ND_{Ar\times(N2,O2)}^{air}d_{N2} - ND_{Ar}^{air}d_{Ar} \qquad (A11)$$

where

$$D_{N2}^{air} = \frac{D_{N2:O2}(C_{N2}D_{Ar:O2} + (1 - C_{N2})D_{N2:Ar})}{S} \qquad (A12)$$

$$D_{N2\times(Ar,O2)}^{air} = \frac{C_{N2}D_{Ar:O2}(D_{N2:O2} - D_{N2:Ar})}{S} \qquad (A13)$$

$$D_{Ar}^{air} = \frac{D_{Ar:O2}(C_{Ar}D_{N2:O2} + (1 - C_{Ar})D_{N2:Ar})}{S} \qquad (A14)$$

$$D_{Ar\times(N2,O2)}^{air} = \frac{C_{Ar}D_{N2:O2}(D_{Ar:O2} - D_{N2:Ar})}{S} \qquad (A15)$$

$$S = C_{N2}D_{Ar:O2} + C_{Ar}D_{N2:O2} + C_{O2}D_{N2:Ar} \qquad (A16)$$

Here $D_{N2}^{air}$ and $D_{Ar}^{air}$ are the effective diffusivities of N$_2$ and Ar in air, while $D_{N2\times(Ar,O2)}^{air}$ and $D_{Ar\times(N2,O2)}^{air}$ reflect ternary cross
interactions, such as the tendency of N$_2$ to be impacted by any process that drives a diffusive flux of Ar.





In air, Ar is a minor gas ($C_{Ar} \ll C_{N2} \sim C_{O2}$) and therefore interactions of N$_2$ with Ar can be neglected in the N$_2$ flux and the diffusive fluxes of N$_2$ and O$_2$ must balance approximately as in the case of a binary mixture of the two gases

$$f_{N2} \approx -ND_{N2}^{air} d_{N2} \tag{A17}$$

$$f_{O2} \approx -f_{N2} \tag{A18}$$

Combining Eqs. (A10) and (A17) with Eq. (A9), yields

$$f_{N2} \approx -ND_{N2}^{air}\left[\nabla C_{N2} + \left(\frac{1}{H_{N2}} - \frac{1}{H_{air}}\right)C_{N2} + \alpha_{N2:O2}^{T}\frac{C_{N2}(1-C_{N2})}{T}\nabla T\right] \tag{A19}$$

$$f_{Ar} \approx -ND_{Ar\times(N2,O2)}^{air}\left[\nabla C_{N2} + \left(\frac{1}{H_{N2}} - \frac{1}{H_{air}}\right)C_{N2} + \alpha_{N2:O2}^{T}\frac{C_{N2}(1-C_{N2})}{T}\nabla T\right]$$
$$- ND_{Ar}^{air}\left[\nabla C_{Ar} + \left(\frac{1}{H_{Ar}} - \frac{1}{H_{air}}\right)C_{Ar} + \alpha_{Ar:air}^{T}\frac{C_{Ar}}{T}\nabla T\right] \tag{A20}$$

where we have replaced the thermal diffusion ratio $k_i^T$ with the better empirically constrained thermal diffusion factor $\alpha_i^T \equiv$

$\frac{k_i^T}{C_i C_j}$ defined as such only in binary mixtures. Therefore, $k_{N2}^T \approx \alpha_{N2:O2}^T C_{N2}C_{O2}$ and $k_{Ar}^T \approx \alpha_{Ar:air}^T C_{Ar}$. Table A2 presents rough estimates of the magnitudes of the terms in Eqs. (A19) and (A20) in the stratosphere, showing that the thermal diffusion and cross-diffusion terms involving $D_{Ar\times(N2,O2)}^{air}$ are at least two orders of magnitude smaller than the remaining terms. Neglecting these smaller terms yields the governing Eq. (4) used in our model simulation

$$f_i \approx -ND_i\left[\left(\frac{1}{H_i} - \frac{1}{H_{air}}\right)C_i\right] = -ND_i\Delta M_i\frac{g}{RT}C_i \tag{A21}$$

Equation (A21) is equally valid for trace gases such as Ar and major gases N$_2$ and O$_2$ when the appropriate diffusivities given

by Eqs. (A12) and (A14) are used. In our case $D_{N2} \equiv D_{N2}^{air} = D_{O2}^{air} \approx D_{N2:O2}$ for N$_2$ and $D_{Ar} \equiv D_{Ar}^{air} \approx \frac{D_{Ar:O2}D_{N2:Ar}}{C_{N2}D_{Ar:O2}+C_{O2}D_{N2:Ar}}$ for Ar.

## 9. Appendix C: Calculating δ(Ar/N₂) from δ(GST)

The conservation equation of gas $i$ with mixing ratio $C_i$ accounting for advection (1st term RHS), eddy mixing (2nd term RHS), and molecular diffusion (3rd term RHS, using the simplified Eq. (A21)) is given by

$$\frac{\partial}{\partial t}(C_i) = -\vec{u}\cdot\vec{\nabla}[C_i] + \vec{\nabla}\cdot\left[\boldsymbol{D}_e\vec{\nabla}C_i\right] - \frac{\partial}{\partial z}\left[D_i\Delta M_i\frac{g}{RT}C_i\right] \tag{B1}$$



where $\vec{u}$ is the velocity vector (m s$^{-1}$), $\boldsymbol{D}_e$ is the eddy diffusivity tensor (m$^2$ s$^{-1}$), $D_i$ is the molecular diffusivity of species $i$ in air (i.e., $D_i^{air}$ m$^2$ s$^{-1}$), $\Delta M_i$ the molecular mass difference to air (kg mol$^{-1}$), $g$ the gravitational acceleration, $R$ is the fundamental gas constant (J K$^{-1}$ mol$^{-1}$), and $T$ is temperature (K). "$\vec{\nabla}\cdot$" represents the divergence operator. $\vec{u}, \boldsymbol{D}_e, D_i, N$, and $T$ depend on x,y,z, and t and the largest gradients for these variables generally occur in the vertical direction.

Dividing Eq. (B1) by a reference value $C_{i,0}$, separating $C_i$ into a constant and a perturbation component (i.e., $\frac{C_i}{C_{i,0}} = 1 + \frac{C_i'}{C_{i,0}} = $

$1 + \delta_i$), and using the chain rule yields

$$\frac{\partial}{\partial t}(\delta_i) = -\vec{u}\cdot\vec{\nabla}[\delta_i] + \vec{\nabla}\cdot\left[\boldsymbol{D}_e\vec{\nabla}\delta_i\right] - D_i\Delta M_i \frac{g}{RT}\frac{\partial}{\partial z}[\delta_i] + (1+\delta_i)\frac{\partial}{\partial z}\left[D_i\Delta M_i \frac{g}{RT}\right] \qquad (B2)$$

To simplify Eq. (B2), we assume that the perturbations in the mixing ratio are small (i.e., $\delta_i \ll 1$). Furthermore, we assume that $U + \frac{D_e}{L} \gg \frac{D_i}{\Delta H_i}$ ($U$ is a characteristic velocity scale, $D_e$ is a characteristic eddy diffusivity, and $L$ is a characteristic length

scale; $\Delta H_i = \frac{RT}{\Delta M_i g}$) or equivalently stated in terms of the Peclet number $Pe \sim \frac{U\Delta H_i + D\frac{\Delta H_i}{L}}{D_i} \gg 1$. We estimate typical values of the vertical Peclet number in the stratosphere to be between 1000 to 10000 based on the height (~20 km) and turnover time (~5

years) of the stratosphere and the range of molecular diffusivities given in Table A1. Under these conditions, the third term on the RHS can be eliminated because it is $O(Pe)$ smaller than the first and second terms and because it is $O(\delta_i)$ smaller than the fourth term. Thus, we obtain

$$\frac{\partial}{\partial t}(\delta_i) \approx -\vec{u}\cdot\vec{\nabla}[\delta_i] + \vec{\nabla}\cdot\left[\boldsymbol{D}_e\vec{\nabla}\delta_i\right] - \frac{\partial}{\partial z}\left[D_i\Delta M_i \frac{g}{RT}\right] \qquad (B3)$$

Where the last term approximates the vertical divergence of the gravitational flux. At the top of the atmosphere and Earth's surface, the gravitational flux abruptly vanishes, and its divergence becomes large. If its divergence is small in the interior, Eq.

(B4) can be conceptually interpreted as an advection-diffusion problem with large apparent sources or sinks at the bottom and top boundary due to diffusive flux divergence. The steady-state solution to Eq. (B3) for a tracer $\delta_i$ with no additional sources and sinks (other than the apparent sources from diffusive flux divergence) yields the steady-state profile for species $i$. This solution can be scaled to yield the steady-state solution for any other tracer $\delta_j$ as

$$\delta_j \approx \frac{D_j\Delta M_j}{D_i\Delta M_i}\delta_i \qquad (B4)$$

as can easily be validated by solving Eq. (B4) for $\delta_i$ and substitution into Eq. (B3). Note that $\frac{D_j\Delta M_j}{D_i\Delta M_i}$ does not depend on x,y,

and z.

Furthermore, ratios of passive tracers $k$ and $l$ can be calculated directly from $\delta_i$ by recognizing that $\delta_{k/l} \approx \delta_k - \delta_l$ for $\delta_k + \delta_l \ll 1$. Hence, we obtain the general version of Eq. (6) in the main text



$$\delta_{k/l} \approx \ \delta_k - \delta_l \approx \frac{D_k^{air}\Delta M_k - D_l^{air}\Delta M_l}{D_i \Delta M_i}\delta_i. \tag{B5}$$

## 10. Author contributions

MPC, WF and CW set up and ran the TOMCAT model simulations. EJM, JDB, BBS, and SCW collected samples and curated
data. BB carried out the data analysis with important input from MPC, RFK, SCW, ML, and EJM. BB prepared the manuscript
with contributions from all co-authors.

## 11. Competing Interests

The authors declare that they have no conflict of interest.

## 12. Acknowledgements

We would like to thank the pilots and crew of the GV and DC-8 research aircraft as well as NCAR and NASA project managers,
field support staff, and logistics experts that were integral to the success of the ORCAS, HIPPO and ATom campaigns. We
are grateful to Andy Watt at NCAR who helped prepared the Medusa sampler for the field and assisted with its operation. Bill
Paplawsky, Sara Afshar, Shane Clark, Stephen Walker and Adam Cox facilitated sample analysis, gas cylinder preparation,
data management and performed instrument maintenance at Scripps Institution of Oceanography (SIO). $CO_2$ flask analysis at
SIO is supported by NASA grant NNX17AE74G. Special thanks go to Eric Hintsa, Fred Moore, Jim Elkins, Mark Zondlo,
Stuart Beaton, Minghui Diao, Glenn Diskin, Glen Sachse, Joshua DiGangi, John Nowak, James Flynn, Sergio Alvarez,
Kathryn McKain, and Colm Sweeney for sharing $O_3$, $N_2O$, $H_2O$, CO and $CH_4$ data used in this manuscript.

For VCSEL hygrometer, field support, laboratory calibrations and QA/QC were provided by M. Diao, J. DiGangi, M. Zondlo
and S. Beaton in HIPPO; S. Beaton provided field support and M. Diao provided laboratory calibrations and QA/QC in
ORCAS.

This material is based upon work supported by the National Center for Atmospheric Research (NCAR), which is a major
facility sponsored by the National Science Foundation (NSF) under Cooperative Agreement No. 1852977. The HIPPO
program was supported by NSF grants ATM-0628575, ATM-0628519 and ATM-0628388 to Harvard University, University
of California (San Diego), NCAR, and University of Colorado/CIRES. The primary ORCAS program was supported by NSF
Polar Programs Grants 1501993, 1501997, 1501292, 1502301, and 1543457 to NCAR, SIO, University of Colorado/CIRES,
and the University of Michigan. The ATom program was supported by the NASA grant NNX15AJ23G. Medusa sampling on
ATom was supported NSF grants AGS-1547797 and AGS-1623748 to University of California (San Diego) and NCAR. The



TOMCAT simulations were performed on the UK Archer and University of Leeds ARC HPC facilities. The modelling work in Leeds was supported by the NERC SISLAC grant NE/R001782/1. MPC is also supported by a Royal Society Wolfson Merit award.

We thank Dimitry Belikov, Satoshi Sugawara, Shigeyuki Ishidoya and colleagues for sharing the NIES TM modelling results and stratospheric observations from the balloon sampling-system with us.

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





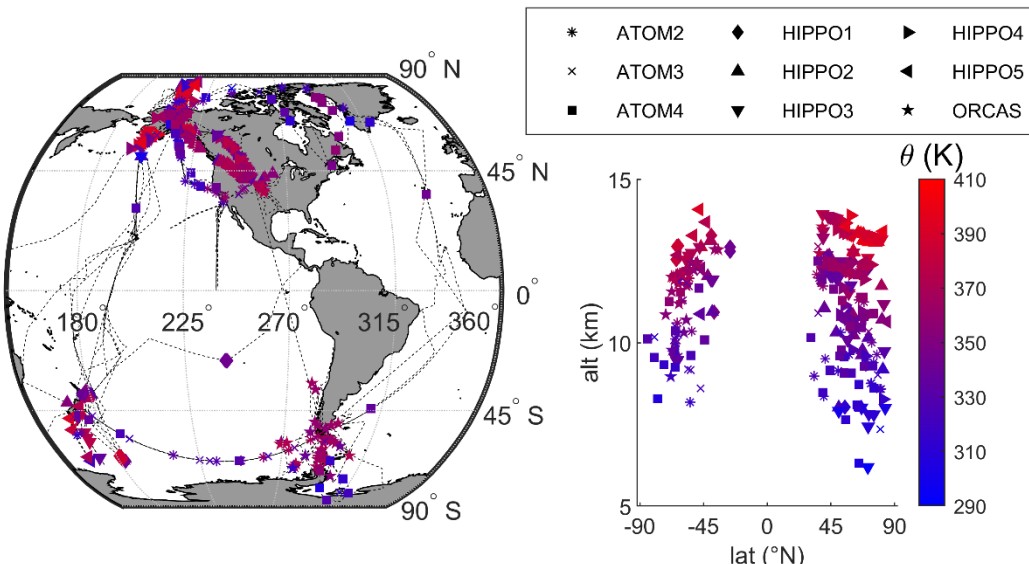

**Figure 1. Horizontal (left) and vertical (right) distribution of airborne flask sample locations identified as being of stratospheric origin (see text). Thin dashed black lines on the map illustrate the flight tracks of all 9 camaigns. Symbols indicate the campaigns during which the stratospheric samples were collected, and colours show the potential temperature at which the sample was taken.**

660

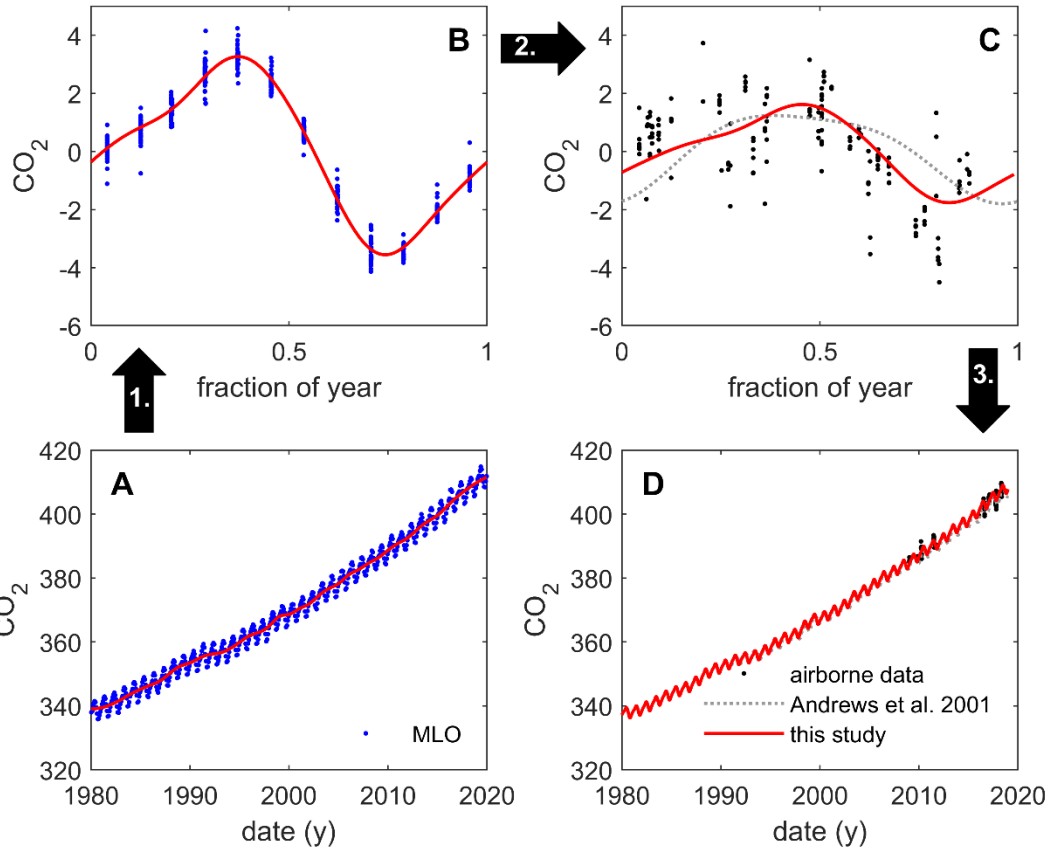

**Figure 2. Illustration of the major steps in deriving a tropical upper tropospheric reference time series. Panel (a) shows the monthly mean surface CO₂ concentrations at Mauna Loa (MLO) (Keeling et al., 2001) with a stiff spline trend shown in red. Panel (b) shows a fit of the mean seasonal cycle at MLO compared to detrended observations from 1980–2019. Panel (c) shows the 1-month-lagged seasonal cycle (red line) rescaled in amplitude to match detrended airborne observations in the equatorial upper troposphere ($20°S < $ lat $ < 20°N$, alt $>8$ km, black). The seasonal cycle derived by Andrews et al. (2001) is presented as a dotted grey line for reference. Panel (d) shows the resulting upper tropospheric mean reference time series (TRTS, red) used for CO₂ in the age of air algorithm with the airborne observations from the tropical upper troposphere (black dots) and the time series (grey dotted line) of Andrews et al. (2001).**

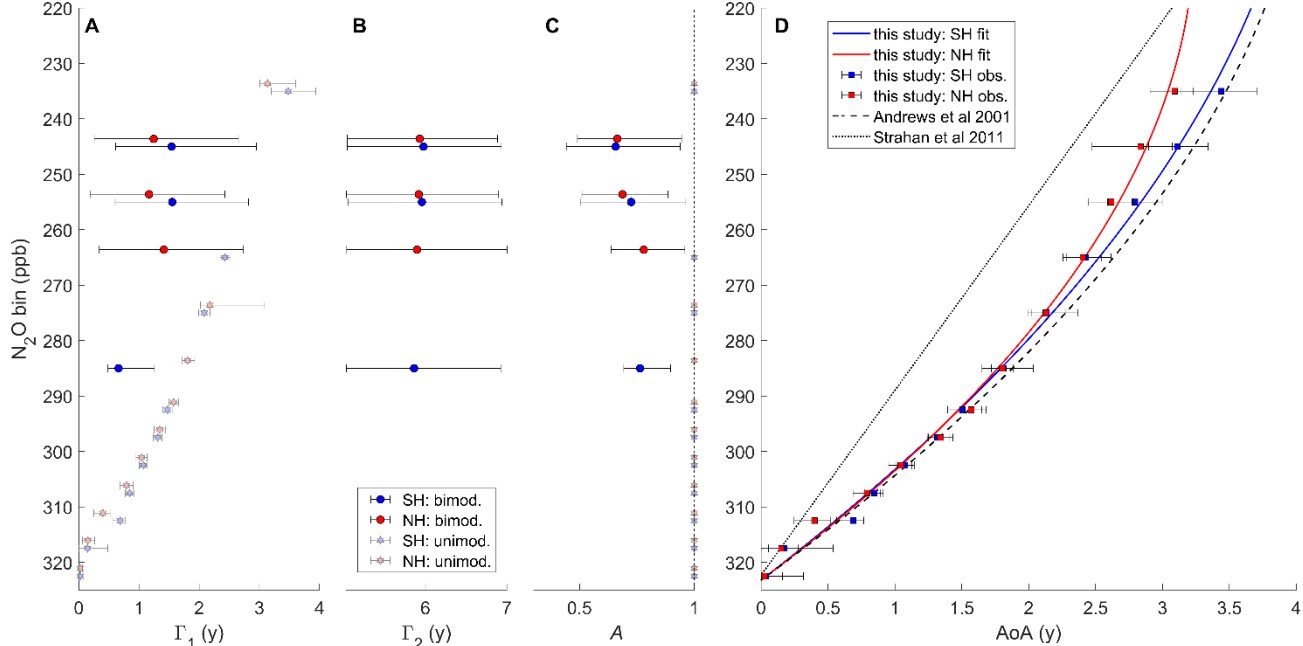

**Figure 3. Hemisphere-specific properties of the age spectra in different $N_2O$ bins from the 10s-averaged airborne observations estimated by Markov Chain Monte Carlo. Panel A-C show the value and 95% confidence interval of $\Gamma_1$, $\Gamma_2$, and $A$ in each bin for either a unimodal (faint blue and faint red stars) or bimodal (blue and red circles) age spectrum depending on which type was preferentially selected by the algorithm. $N_2O$ values are offset by 1.4 ppb for all Northern Hemisphere estimates (NH, red) for visual clarity in panel A-C. Panel D shows the mean AoA and confidence interval of the age spectra ensemble in each bin (i.e., unimodal and bimodal together). These data are fit by a quadratic polynomial for each hemisphere with a fixed y-intercept (NH AoA = -0.0002361 (323.23 - $N_2O$)$^2$ + 0.05530 (323.23 - $N_2O$); SH AoA = -0.0001754 (323.23 - $N_2O$)$^2$ + 0.05359 (323.23 - $N_2O$)). Previous relationships published by Andrews et al. (2001) and Strahan et al. (2011) are given as a dashed and dotted line for reference. Y-intercept values of the previously published $N_2O$-AoA calibrations have been updated to reflect the gradual increase in tropospheric $N_2O$ between 1997 and the new reference year 2009.**



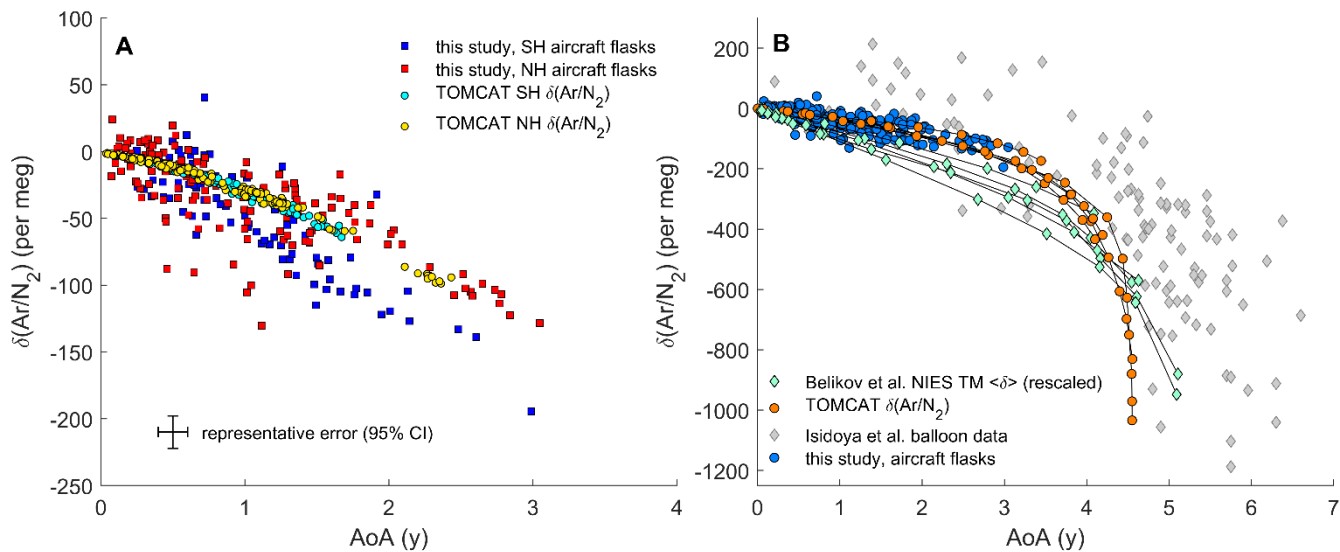


**Figure 4. Comparison of age of air (AoA) and gravitational settling (GS) of δ(Ar/N₂) in the models and observations. Panel (a) shows observations from the Northern (red squares) and Southern Hemisphere (blue squares) together with TOMCAT model output (SH: cyan circles; NH: yellow circles) selected from the closest model grid box in time and space. Because uncertainties for δ(Ar/N₂) and AoA are similar for all stratospheric samples, representative error bars (95% confidence interval) are shown at an arbitrary AoA**

**of ~0.5 years. δ(Ar/N₂) is normalized to yield a delta value of zero in the equatorial free troposphere. In panel (b), observations from airborne campaigns (this study) and the balloon sampling system (Ishidoya et al., 2008, 2013, 2018; Sugawara et al., 2018) are plotted with the AoA-GS relationship observed in TOMCAT and the NIES TM (Belikov et al., 2019) as lines for each of the latitude bands shown in Fig. 5 and points for different altitude bins between 10 and 35 km. To yield an equivalent estimate of δ(Ar/N₂), δ(¹³CO₂/¹²CO₂) results from the NIES TM (Belikov et al., 2019) have been rescaled according to Eq. (5) and offset by ~10 per meg**

**to account for the different tropospheric reference region in the definition of δ.**

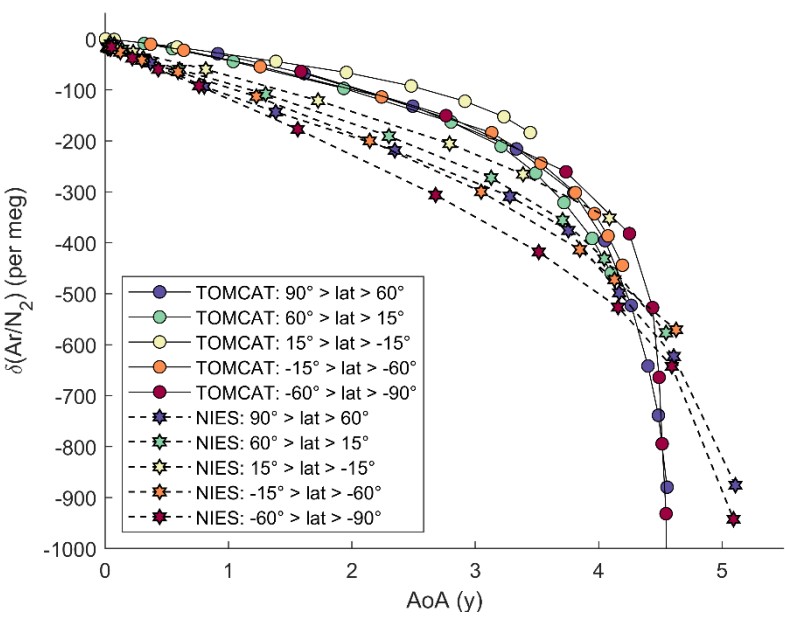


**Figure 5. Comparison of the annual and zonal mean age of air (AoA) relationship to gravitational settling simulated in TOMCAT and the NIES TM. The relationship is plotted as lines for the latitude bands indicated by marker colour. For the NIES TM, markers show the vertical profile using all grid boxes available between ~10-35 km. For TOMCAT, markers instead correspond to binned altitude bands between 10 and 35 km with a spacing of 2.5 km because of the finer vertical resolution of the model. To yield an**
**equivalent estimate of δ(Ar/N₂), δ(¹³CO₂/¹²CO₂) results from the NIES TM (Belikov et al., 2019) have been rescaled according to Eq. (5) and offset by ~10 per meg to account for the different tropospheric reference region in the definition of δ.**





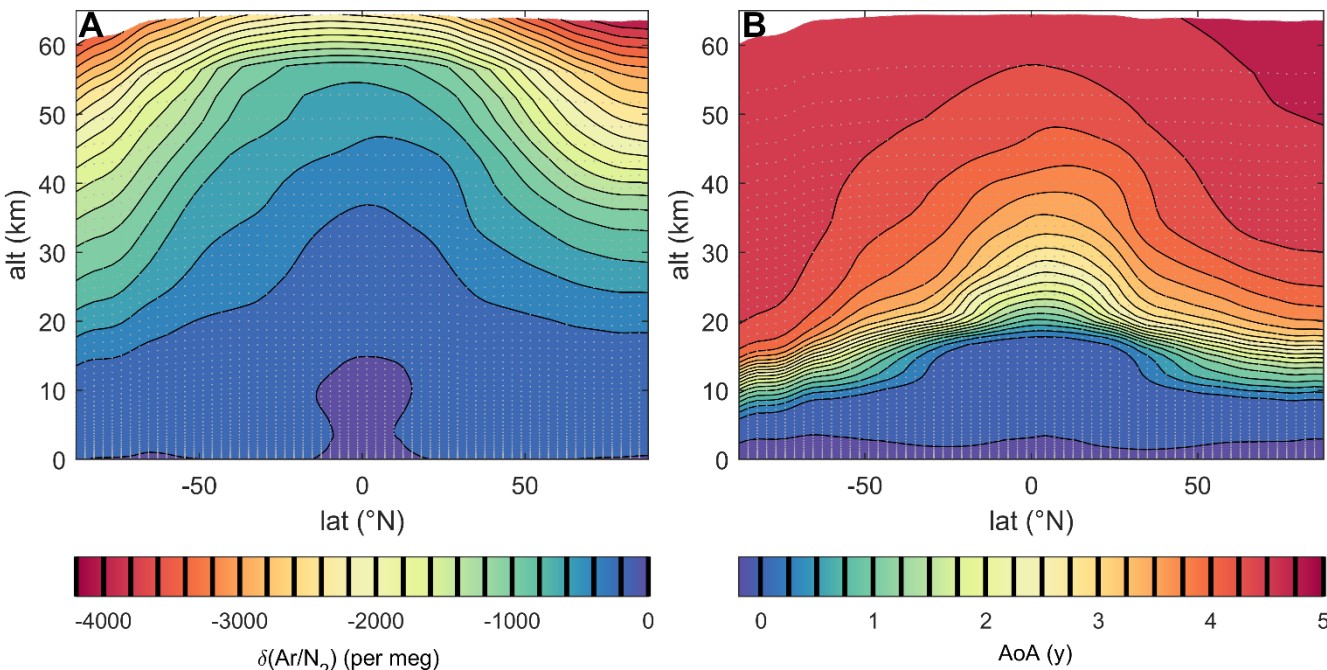

**Figure 6. Annual and zonal mean of δ(Ar/N₂) (panel A) and age of air (AoA) (panel B) simulated by TOMCAT. Values of the contour lines are shown as black vertical lines on the colour bar. Grey dots indicate the centre of a grid boxes in TOMCAT. A video of this figure highlighting the natural variability in monthly-mean values is available online with this manuscript.**




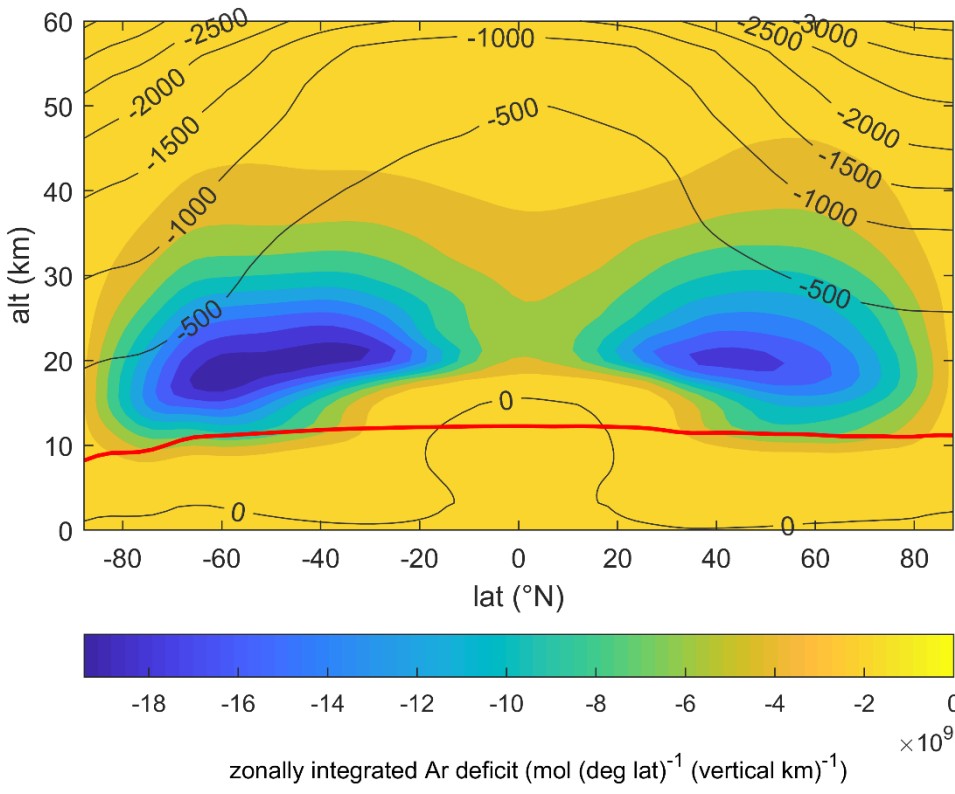

**Figure 7. Colormap of annual mean, zonally integrated Ar deficit in TOMCAT. Overlain black contour lines show δ(Ar/N₂) (per meg). The red solid line highlights the mean position of the 200 hPa isobar.**






**Table A1. Molecular diffusion volumes (Reid et al., 1987) and masses used in this study**

| Chemical species | Molecular mass (g mol$^{-1}$) | Molecular diffusion volume |
|---|---|---|
| Ar | 40 | 16.2 |
| N$_2$ | 28 | 18.5 |
| O$_2$ | 32 | 16.3 |
| air | 28.95 | 19.7 |




**Table A2. Characteristic values of variables in Eqs. (A14) and (A15)**

| Variable/term | magnitude at ~20 km | magnitude at ~35 km | units | Notes or Eq. number |
|---|---|---|---|---|
| $P$ | 50 | 5 | mb | |
| $T$ | 215 | 250 | K | |
| $\dfrac{\partial T}{\partial z}$ | 5e-04 | 3e-03 | K m$^{-1}$ | |
| $H_{N2}$ | 6.51e+03 | 7.57e+03 | m | |
| $H_{Ar}$ | 4.56e+03 | 5.30e+03 | m | |
| $H_{air}$ | 6.29e+03 | 7.32e+03 | m | |
| $\alpha_{N2:O2}^{T}$ | <1.8e-02 | | - | 1 |
| $\alpha_{Ar:air}^{T}$ | ~5.6e-02 | ~6.6e-02 | - | 2 |
| $C_{N2}$ | 78.09 | | % | |
| $C_{O2}$ | 20.95 | | % | |
| $C_{Ar}$ | 0.93 | | % | |
| $\dfrac{\partial C_{N2}}{\partial z}$ | 3.3e-09 | 7.2e-09 | m$^{-1}$ | 3 |
| $\dfrac{\partial C_{Ar}}{\partial z}$ | 3.3e-10 | 7.4e-10 | m$^{-1}$ | 3 |
| $D_{N2}^{air}$ | 2.3e-04 | 3.0e-03 | m$^2$s$^{-1}$ | |
| $D_{Ar\times(N2,O2)}^{air}$ | 9.8e-09 | 1.3e-07 | m$^2$s$^{-1}$ | |
| $D_{Ar}^{air}$ | 2.2e-04 | 2.9e-03 | m$^2$s$^{-1}$ | |
| $D_{N2}^{air}\dfrac{\partial C_{N2}}{\partial z}$ | 6.3e-13 | 8.2e-12 | m s$^{-1}$ | A19 |
| $D_{N2}^{air}\left(\dfrac{1}{H_{N2}}-\dfrac{1}{H_{air}}\right)C_{N2}$ | 9.6e-10 | 1.1e-08 | m s$^{-1}$ | A19 |
| $D_{N2}^{air}\alpha_{N2:O2}^{T}\dfrac{C_{N2}(1-C_{N2})}{T}\dfrac{\partial T}{\partial z}$ | <1.6e-12 | <1.1e-10 | m s$^{-1}$ | A19 |
| $D_{Ar\times(N2,O2)}^{air}\dfrac{\partial C_{N2}}{\partial z}$ | 2.6e-17 | 3.4e-16 | m s$^{-1}$ | A20 |
| $D_{Ar\times(N2,O2)}^{air}\left(\dfrac{1}{H_{N2}}-\dfrac{1}{H_{air}}\right)C_{N2}$ | 4.0e-14 | 4.5e-13 | m s$^{-1}$ | A20 |
| $D_{Ar\times(N2,O2)}^{air}\alpha_{N2:O2}^{T}\dfrac{C_{N2}(1-C_{N2})}{T}\dfrac{\partial T}{\partial z}$ | <6.8e-17 | <4.6e-15 | m s$^{-1}$ | A20 |
| $D_{Ar}^{air}\dfrac{\partial C_{Ar}}{\partial z}$ | 7.8e-14 | 9.7e-13 | m s$^{-1}$ | A20 |
| $D_{Ar}^{air}\left(\dfrac{1}{H_{Ar}}-\dfrac{1}{H_{air}}\right)C_{Ar}$ | 1.3e-10 | 1.4e-09 | m s$^{-1}$ | A20 |
| $D_{Ar}^{air}\alpha_{Ar:air}^{T}\dfrac{C_{Ar}}{T}\dfrac{\partial T}{\partial z}$ | ~2.7-13 | ~2.1e-11 | m s$^{-1}$ | A20 |

[1] 1.8e-02 observed at 293 K (Waldmann, 1947)

[2] assuming $C_{Ar} \approx \dfrac{N_{Ar}}{N_{N2}}$

[3] estimated from TOMCAT results and observations