# Peer review of "Gravitational separation of $Ar/N_2$ and age of air in the lowermost stratosphere in airborne observations and a chemical transport model"

_Atmospheric Chemistry and Physics, 2020_

## Referee Comment (RC1) · Anonymous Referee #1 · 25 Mar 2020

Comments on acp-2020-95

**General comments**

This paper extends the investigation in earlier papers of the gravitational separation (hereafter GS) and its relationship with age of air in the stratosphere. What is novel about this paper is its focus on availability of $Ar/N_2$ ratio as an independent age tracer. This is a new idea because $Ar/N_2$ ratio is totally different from "clock" tracers that are often used for age observations based on monotonic increase trends in troposphere. Moreover, $Ar/N_2$ has an advantage as an age tracer because it will be free from unfavorable influences of the propagation of seasonal variations and chemical processes on age estimation. In the previous studies, the relationships between age and GS have been examined, but observation data was quite limited and showed somewhat large scatter and there was no idea to use it for age tracer. This study shows tight correlations between AoA and GS by using aircraft data although it is limited to the lower stratosphere, which is supported by suitable 3-D model simulations. Another good point is sufficient theoretical development for the molecular diffusion process in atmosphere. In a sense, the atmospheric GS is somewhat new science, although the theory of molecular diffusion itself is well established. The authors presented a careful consideration of the molecular diffusion process in a ternary mixture by reviewing the theory from the basics and derived an unambiguous and generalized definition of the molecular diffusion coefficient as well as a simple method of GS simulation. These results will be very useful for future GS research, especially for model study.

This paper is worthy of publication in principle, subject to attention to a few issues as follows.

**Specific comments**

(1) As described in section 2.1, inlet fractionation is problematic for $Ar/N_2$ data obtained by airborne observations. The authors eliminated some data considering apparent biases or bad quality. I recommend that a little more details of data quality check or a criterion will be explained. The reason I care about this is that we can see clear difference of AoA-GS relationship between SH and NH (Fig. 4 and Lines

236-240). Is there a possibility that the data is still affected by the inlet fractionation and/or sample deterioration?

(2) Needless to say, it is useful to show the differences of vertical distributions between the model results and the observation for the model studies. Actually, AoA calculated by TOMCAT has been compared with observations in the previous studies (e.g. Chabrillat et al., 2018). However, the comparisons with the observation results are only shown as the AoA-GS relationships in this paper and we cannot know how TOMCAT reproduces vertical distributions of GS. It would be better to show a direct comparison with the balloon GS data. Particularly, the authors emphasize that TOMCAT includes GS enhancement in the upper stratosphere and mesosphere, which is superior to the previous model in this respect. This advantage will bring a significant improvement especially in the vertical structure of GS. That is one of the reasons why it is recommended to compare with the observations for not only the AoA-GS relationships in the lower stratosphere but also the vertical distributions observed by balloons.

(3) The authors hypothesize that an adequate representation of the mesosphere in models is critical for the AoA-GS relationship (Line 257). In fact, TOMCAT has a great advantage, because the top of model atmosphere is much higher than that in previous model study. However, I cannot fully understand that neglecting some components of molecular diffusion processes is really adequate even in mesosphere. In this model study, the molecular diffusion processes arisen from the concentration gradient and the temperature gradient are neglected. As summarized in Table A2, those components will be quite smaller than the pressure gradient term and negligible below around 35 km. However, the molecular diffusion coefficient becomes large in the upper stratosphere and mesosphere, which means that Peclet number will be small and thus component of concentration gradient (corresponds to the 3rd term RHS of eq. B2) is not negligible. In such a case, the assumption of the steady-state solution (eq. B4) is also not true. In this connection, magnitude of GS shown in Fig. 6 is much smaller than that of Fig.4 in Ishidoya et al. (2013) in the mid-stratosphere over high latitudes where the mesospheric GS will strongly influence due to the downward transport. It would be better to show that the influences of ignoring some terms and the assumption of a steady state are small even in the upper stratosphere and mesosphere.

(4) Another question arises also about the steady-state assumption concerning the seasonal variations. We can find that large seasonal variations of $Ar/N_2$ ratio occur in stratosphere and mesosphere as shown in supplement movie (Lines 249-251). This simply means that the time derivative of $\delta$ is not zero in eq. B3 and that there should be some restrictions for the approximation (eq. B4) to be applied. I just feel that an effect of the diffusive separation in the non-steady state will be small for $Ar/N_2$, because difference of the molecular diffusion coefficients between Ar and $N_2$ is not so large compared with those of heavy noble gases. However I don't know how far this steady-state assumption can be generalized.

**Technical comments**

Minor comments are listed below.

(L-109): "...are available from the START-08 campaign on the NSF/NCAR GV, but we have not used these here because the $\delta(Ar/N_2)$ data quality is considerably worse. "
What is the reason why the quality of START-08 data is worse?

(Fig. 1): Some symbols (rotated triangles?) do not seem to match those in the legend.

(Eq. 3): "$\lambda$" in the 2nd term of RHS should be $\lambda_2$. $G(t'|(\Gamma, \lambda))$ should be $G(t'|(\Gamma_1, \Gamma_2, \lambda_1, \lambda_2))$, to be exact.

(L-152): "$\Gamma_1$, $\Gamma_1$, $\lambda_1$ and $\lambda_2$ are... " should be "$\Gamma_1$, $\Gamma_2$, $\lambda_1$ and $\lambda_2$ are ...."

(L-179): "(ii)" should be "(iii)".

(Section 3.1): Parameters $\Gamma_1$, $\Gamma_2$, and A are shown in Fig. 3 and discussed in detail. But there is no description how the shape parameters $\lambda_1$ and $\lambda_2$ of the inverse-Gaussian distribution were as a result. It would be better to give us information about widths ($\Delta$) or values of $\gamma(=\Delta^2/\Gamma)$ of age spectra obtained by this method.

(L-183 to Eq. 5): "....atmosphere $\delta_{GST}$ with a molecular mass 1 amu greater than that of air ..."

GST seems to be a kind of virtual tracer. It may be better to mention how the molecular diffusion volume was defined for $D_{GST}$.

(L-236 to 240): Again, with this AoA-GS plots alone, it is somewhat difficult to understand the difference between NH and SH. Vertical distributions of AoA and GS may be helpful.

(L-276 and Fig. S2): "...the small seasonal cycle amplitude of $\delta(Ar/N2)$ <6 per meg..."

Certainly the average seasonal amplitude seems to be small, but we can see large scatters. Fluctuations of $Ar/N_2$ ratio in short time will be partly smoothed by the mixing process during the upward transport from the tropical upper troposphere to the lower stratosphere via TTL. Thus, it may not be a big obstacle to AoA estimation from $Ar/N_2$. Problem is, rather, that this scatter is real atmospheric signal or not.

(Eq. 7 in Appendix A): This number should be something like "(A1)". Also please check eq. numbers in Appendix B and C.

(L-411):"$\Delta M_i$ the molecular mass difference to air "

$\Delta M_i$ appears in eq. (A21) for the first time.

(Eqs. B1, B2, and B3): Please check the signs (plus/minus) of the molecular diffusion terms in these equations. In the conservation equation, term of the flux divergence (3rd term RHS in eq. B1) should be a form like $-\nabla \cdot [f]$ if we put it on RHS, just the same as the eddy diffusion term. The 3rd and 4th terms of RHS in eq. B2 might have the same signs after using the chain rule. The 3rd term of RHS in eq. B3 might have the same sign with 4th term of RHS in eq. B2 after eliminating a small term.

(L-418): "D" in the expression of Peclet number should be "$D_e$".

(L-432): "Eq. (6)" should be Eq. (5).

---

## Referee Comment (RC2) · Anonymous Referee #2 · 27 Mar 2020

This paper uses measured and simulated ratios of Ar/N2 to deduce gravitational separation in the stratosphere and their relationship to the stratospheric age of air. The Ar/N2 relationships to AoA in this study are relatively compact compared to the previous Belikov et al. study on gravitational separation and the agreement between the observations and model relationships are improved. This suggests that gravitational separation of Ar/N2, and also potentially the ratios of heavier noble gases, could be useful as an additional diagnostic of transport. The measurements, model runs and methodology are well described and the improvement in the treatment of gravitational

separation in both the measurements and model compared to previous studies is an important step forward.

My main concern with this study is in the calculation of age of air from the measurements. The technique used, although rigorous, is based on assumptions that are not consistent with our current understanding of the shape of age spectra and the transport of air into the lowermost stratosphere. The technique and assumptions are based on a study that is nearly two decades old and there have been numerous studies since that time refining our understanding of age spectra and how they can be inferred from measurements. The results may not change substantially by using more realistic age spectra and transport assumptions but that isn't entirely clear. Regardless, the more recent studies on this topic and the newer techniques should be acknowledged and shown to be in agreement with, or ideally replace, the results from the technique used in this study. Specific references and more detail on this topic are included below.

The results of this study are significant and relevant to ACP so I recommend publication after modification of the age of air calculation and consideration of the specific comments below.

Specific comments:

Line 35: should add 'however' before 'observational'

Lines 40-41: should add references here to studies that have used tracers to infer age spectra (Andrews et al., JGR, 1999; Schoeberl et al., JGR, 2005; Hauck et al., ACP, 2019; Podglajen and Ploeger, ACP, 2019).

Line 94: did you mean 'CO2' instead of 'CO' or should 'CO2' be added here?

Line 97: 'sample flask i to sample flask i+1'

Line 108: change 'as a combined result of' to 'due to'

Line 113: 'used' instead of 'use'

Section 2.2: This section is my main concern with the paper as I mentioned above. I have two issues: (1) the assumed shape of the age spectra and (2) the tropical single-entry point assumption for the lowermost stratosphere. The first issue, the shape of stratospheric age spectra has been shown in a number of studies derived from various model and reanalysis output based on trajectories or boundary impulse functions (Reithmeier et al., Clim. Dyn., 2008; Li et al., JGR, 2012; Diallo et al., ACP, 2012; Ray et al., JGR, 2014; Ploeger and Birner, ACP, 2016; Podglajen and Ploeger, ACP, 2019; Hauck et al., ACP, 2019). In these studies, it is clear that the dominant mode of variability in the age spectra in the lower stratosphere is the seasonal cycle, and for those models with a QBO, that is the second largest mode of variability. None of these results show a peak in the 5-7 year range relative to the 2-5 year range of the spectra. The use of two peak age spectra with one of the peaks in the 5-7 year range is not realistic for any part of the stratosphere and especially not in the lower stratosphere where the seasonal cycle is so large and the 5-7 year portion of the age spectra has much smaller values (often by an order of magnitude) compared to those at ages less than 5 years. The lack of reference in the manuscript to any of the papers listed above and the use of an age spectra shape that is not consistent with any of them is a clear deficiency.

The use of a single trace gas, such as $CO_2$, can only reveal a limited amount of the age spectra. It has been shown by Hauck et al. (2019) that at least 5-10 tracers with a range of lifetimes are necessary to resolve the important features and the true shape of the age spectra. The Andrews et al. (1999, 2001) studies were important steps at the time in our understanding of how trace gas observations could be used to infer features of the age spectra. But now that we have a detailed understanding of age spectra from models and reanalysis it isn't appropriate to use age spectra defined only by the fit to $CO_2$.

The second issue is related to the transport of air into the lowermost stratosphere (LMS) where most of the measurements in this study were taken. Many studies have shown that the LMS consists of a seasonally varying mixture of tropospheric and strato-

spheric air with some portion of the tropospheric air originating from the extratropics or tropical air that has bypassed the tropical tropopause (e.g. Ray et al., JGR, 1999; Hoor et al., ACP, 2004; Olsen et al., JGR, 2004; Boenisch et al., ACP, 2009; Skerlak et al., ACP, 2014). This precludes the use of a single-entry tropical tropopause time series to derive age of air in the LMS.

I highly recommend a technique considering the above points be used here. Even if the resulting mean age values do not differ significantly, the technique would be consistent with our current understanding of transport in the lower stratosphere.

Line 152: Gamma1 is repeated, the second one should be Gamma2.

Line 179: the second '(ii)' should be '(iii)'

Line 203: 'relationships'

Line 262: 'descends' instead of 'is sinking'

Line 328: Is the Ar/N2 ratio 'mostly unaffected by seasonality' throughout the stratosphere?

Lines 330-331: It's not so much that changes in AoA in the upper stratosphere and mesosphere are difficult to resolve from transient tracers but more that AoA becomes nearly constant in this region as you show in Figure 6. So AoA loses sensitivity in this region where gravitational separation gains sensitivity. This is a difficult region to make measurements of any kind so it may also be challenging to obtain enough measurements of noble gases to monitor gravitational separation.

---

## Author Comment (AC1) · 16 May 2020

Please find the response to reviewers including a marked-up version of the manuscript in the document attached to this comment.

Please also note the supplement to this comment:
https://www.atmos-chem-phys-discuss.net/acp-2020-95/acp-2020-95-AC1-supplement.pdf

---

## Author Response (AR2)

**Response to Reviewers - Acp-2020-95**

We thank anonymous Reviewers 1 and 2 for their insightful comments and discussion of the manuscript. We provide an updated document, including updated figures, that incorporates the suggested changes and address specific comments below. For select comments, we list line numbers of modified sections of the manuscript and include the most relevant excerpts.

**Comments from Reviewer 1**

(1) As described in section 2.1, inlet fractionation is problematic for Ar/N2 data obtained by airborne observations. The authors eliminated some data considering apparent biases or bad quality. I recommend that a little more details of data quality check or a criterion will be explained. The reason I care about this is that we can see clear difference of AoA-GS relationship between SH and NH (Fig. 4 and Lines 236-240). Is there a possibility that the data is still affected by the inlet fractionation and/or sample deterioration?

While data within a single campaign are most comparable because the position, aircraft type and parts of the sampling equipment evolved between campaigns, some offsets exist between campaigns. We exclude Atom 1 due to obvious artifacts relating to both the inlet employed and the position of the inlet over the wing. The same inlet was used during ORCAS and ATom1, but in the case of ORCAS the positioning of the inlet on the fuselage of the aircraft eliminated some of the larger artifacts seen in ATom1. We think the ORCAS data are of acceptable quality for publication, but as most of the stratospheric samples from the Southern Hemisphere in our dataset are from ORCAS, we are therefore careful about interpreting the apparent difference between the Southern and Northern Hemisphere in the GS-AoA relationship, until new observations are available to confirm or refute this. This is clarified in the manuscript (line 279-282 and SI).

From manuscript: "However, almost all SH samples with mean AoA older than 1.5 years come only from the ORCAS campaign. This campaigned used a different inlet design than HIPPO and ATom which could cause a small artifactual offset and but the scatter in our observations generally makes it difficult to separate signal from noise for small interhemispheric differences."

From SI: "**$\delta(Ar/N_2)$ residual analysis**

To test the hypothesis that large artifacts or biases in $\delta(Ar/N_2)$ were introduced during flask sample collection, we look for the presence of a pressure dependency of $\delta(Ar/N_2)$ after removing what we have shown is the natural gravitational signal. Pressure serves as a proxy for sampling conditions and aircraft velocity simultaneously here because aircraft velocity also increases systematically with altitude or pressure altitude. A vertical profile of $\delta(Ar/N_2)$ (Fig. S1A) reveals no strong dependency on pressure in the troposphere below about 300 mb. Above ~300 mb, some samples represent stratospheric conditions and therefore have an age of air (AoA) greater than zero and more negative $\delta(Ar/N_2)$. For further analysis, we correct $\delta(Ar/N_2)$ for the natural gravitational signal ($\delta(Ar/N_2)_{grav}$) using observed AoA from each sample and a quadratic fit of the AoA-$\delta(Ar/N_2)$ relationship simulated in TOMCAT ($\delta(Ar/N_2)_{grav} = -8.124\,AoA^2 - 22.73\,AoA + 0.9529$). The result is shown in Figure S1B. Although there is scatter in the corrected data, in particular for HIPPO1–5, no systematic relationship between pressure and gravitationally-corrected $\delta(Ar/N_2)$ is evident, supporting the interpretation that aircraft sampling did not create large biases in $\delta(Ar/N_2)$. While there is some evidence of a pressure dependence in the ORCAS data from the Southern Hemisphere (SH), this could also be the result of an incomplete correction for gravitational fractionation if there are differences in the true AoA-$\delta(Ar/N_2)$ relationship for the SH not captured by TOMCAT. However, stratospheric

samples from ORCAS were almost exclusively taken at the same pressure, so our analysis of the $\delta(Ar/N_2)$-AoA relationship should be robust regardless."

(2) Needless to say, it is useful to show the differences of vertical distributions between the model results and the observation for the model studies. Actually, AoA calculated by TOMCAT has been compared with observations in the previous studies (e.g. Chabrillat et al., 2018). However, the comparisons with the observation results are only shown as the AoA-GS relationships in this paper and we cannot know how TOMCAT reproduces vertical distributions of GS. It would be better to show a direct comparison with the balloon GS data. Particularly, the authors emphasize that TOMCAT includes GS enhancement in the upper stratosphere and mesosphere, which is superior to the previous model in this respect. This advantage will bring a significant improvement especially in the vertical structure of GS. That is one of the reasons why it is recommended to compare with the observations for not only the AoA-GS relationships in the lower stratosphere but also the vertical distributions observed by balloons.

We thank the reviewer for the suggestion. A direct comparison of aircraft-observed and TOMCAT-simulated $\delta(Ar/N_2)$ and AoA in the lowermost stratosphere is shown in Figure S4 which we now reference in the text (lines 265-269). However, a direct comparison of our aircraft observations and modeling results in this case is not a particularly good metric of model performance or data quality because the samples represent air with unusual chemical composition for its environment. These filaments are often small-scale features that cannot be reproduced by TOMCAT with a model resolution of 2.8°. Therefore, simulated AoA in many samples is considerably lower than observed and $\delta(Ar/N_2)$ is correspondingly less fractionated. These effects at least partially cancel when studying the GS-AoA ("tracer-tracer") relationship and make it a more robust metric for model-data comparison.

TOMCAT has previously been validated against a range of AoA observations, as pointed out by the reviewer, and the configuration used for this study has a tendency to slightly underestimate AoA as discussed in the manuscript (line 262). This will also introduce a bias in $\delta(Ar/N_2)$ based on the robust relationship between $\delta(Ar/N_2)$ and AoA. Hence, we believe that a comparison to the available balloon data against pressure would not be particularly instructive and distract from the main message of the paper.

From manuscript: "A comparison of $\delta(Ar/N_2)$ and AoA between flask samples and TOMCAT is shown in Figure S4. However, the direct comparison should not be considered a good test of model performance because our stratospheric composition criteria intentionally target unusually old air in the lowermost stratosphere. For the same reason, profiles of $\delta(Ar/N_2)$ and AoA are not evaluated here. Instead, we suggest the AoA-GS relationship provides a more robust metric for comparison. Overall, our observations validate the implementation of GS for young (< 3 y) ages in TOMCAT."

(3) The authors hypothesize that an adequate representation of the mesosphere in models is critical for the AoA-GS relationship (Line 257). In fact, TOMCAT has a great advantage, because the top of model atmosphere is much higher than that in previous model study. However, I cannot fully understand that neglecting some components of molecular diffusion processes is really adequate even in mesosphere. In this model study, the molecular diffusion processes arisen from the concentration gradient and the temperature gradient are neglected. As summarized in Table A2, those components will be quite smaller than the pressure gradient term and negligible below around 35 km. However, the molecular diffusion coefficient becomes large in the upper stratosphere and mesosphere, which means that Peclet number

will be small and thus component of concentration gradient (corresponds to the 3rd term RHS of eq. B2) is not negligible. In such a case, the assumption of the steady-state solution (eq. B4) is also not true. In this connection, magnitude of GS shown in Fig. 6 is much smaller than that of Fig.4 in Ishidoya et al. (2013) in the mid-stratosphere over high latitudes where the mesospheric GS will strongly influence due to the downward transport. It would be better to show that the influences of ignoring some terms and the assumption of a steady state are small even in the upper stratosphere and mesosphere.

We appreciate the reviewer's comment. Following the suggestion, we have added another column to Table B2 representing mesospheric condition and generally updated the table to reflect extratropical conditions in TOMCAT better (see updated manuscript). The scaling of terms confirms that our approximation of neglecting the thermal diffusion term and mixing by molecular diffusion remains valid because changes in temperature and mixing ratio gradients between the regions are only moderate.

The Peclet number in the mesosphere is still much greater than 1 because not only molecular diffusion but also winds speeds increase with altitude (lines 467-468). This can be shown again by a back-of-the-envelope calculation. AoA is near-uniform in the mesosphere and we can conservatively assume a mesospheric turnover time of 0.5 years and height of 20 km. Comparing that to the molecular diffusivities of $N_2$ and Ar added to Table B2 implies a Peclet number of Pe $\sim$500.

From manuscript:"In the mesosphere, AoA is near-uniform and we conservatively assume a turnover time of 0.5 years and height of 20 km. This implies a Peclet number of $\sim$500 for the mesosphere."

(4) Another question arises also about the steady-state assumption concerning the seasonal variations. We can find that large seasonal variations of Ar/N2 ratio occur in stratosphere and mesosphere as shown in supplement movie (Lines 249-251). This simply means that the time derivative of δ is not zero in eq. B3 and that there should be some restrictions for the approximation (eq. B4) to be applied. I just feel that an effect of the diffusive separation in the non-steady state will be small for Ar/N2, because difference of the molecular diffusion coefficients between Ar and N2 is not so large compared with those of heavy noble gases. However I don't know how far this steady-state assumption can be generalized.

Spatial and time derivatives of $\delta(Ar/N_2)$ are indeed not zero but even in the mesosphere remain considerably smaller than 1 when not given in per meg units. A mesospheric delta value of -5000 per meg implies deviation from a uniform mixing ratio of 0.5% which is still a good approximation.

**Technical comments**
(L-109): "...are available from the START-08 campaign on the NSF/NCAR GV, but we have not used these here because the δ(Ar/N2) data quality is considerably worse. " What is the reason why the quality of START-08 data is worse?

We thank the reviewer for raising this question. START-08 was considered a pre-HIPPO test campaign. It was the first campaign in which the Medusa whole air sampler was used since being repackaged and partially automated following the COBRA and IDEAS campaigns, and the data quality from the campaign, as evident in the much larger scatter of vertical profiles, seems to be worse. During START-08, Medusa sampled from a relatively long polyethylene lined inlet tube, which may have contributed to adverse surface interactions. Subsequent advances in sampling have considerably improved data quality by avoiding surface effects, artifactual fractionation, and analysis problems. Uncertainty and recent advances

in $\delta$(Ar/N$_2$) data quality are now discussed in more detail in a separate section of the methods (lines 87-99) and are also described in Bent (2014).

From manuscript: "Uncertainty in $\delta$(Ar/N$_2$) observations arises from a combination of analytical limitations and artifactual fractionation during sampling. Replicate agreement of surface flasks shows a 1$\sigma$ repeatability of ±6.1 per meg for $\delta$(Ar/N$_2$) but additional scatter in the data may be introduced by small leaks in the Medusa system and thermal or pressure gradient fractionation at the sample intake (Morgan et al., 2020). For further details, see Bent (2014). These effects are challenging to separate from true atmospheric variability and differ between campaigns because sampling strategies have generally improved over time. To constrain uncertainty due to aircraft sampling, we quantify total variability in observations from the presumably fairly homogeneous tropical troposphere between 3-8 km (Fig. S1). Data from the earlier campaigns HIPPO 1–5 show a pooled standard deviation of 24 per meg whereas data from ATom 2–4 yield a pooled standard deviation of 9 per meg, illustrating advances in sampling and sample handling. While no ORCAS samples are available in the tropical troposphere, ORCAS data show similar scatter as ATom 2–4 data between 20°S and 50°S. For all campaigns, the uncertainty is small compared to the stratospheric signal of tens to hundreds per meg shown below. We also show that the stratospheric signal in $\delta$(Ar/N$_2$) is not due to pressure-dependent inlet fractionation by evaluating the residual $\delta$(Ar/N$_2$), which has been corrected for the natural gravitational signal (Fig. S2)."

(Fig. 1): Some symbols (rotated triangles?) do not seem to match those in the legend.

All symbols seem to match in our version of the figure. There are only 4 different 90-degree-rotated triangles pointing in each compass direction.

(Eq. 3): "λ" in the 2nd term of RHS should be λ2. $G(t'|(\Gamma, \lambda))$ should be $G(t'|(\Gamma 1, \Gamma 2, \lambda 1, \lambda 2))$, to be exact.
(L-152): "Γ1, Γ1, λ1 and λ2 are... " should be "Γ1, Γ2, λ1 and λ2 are ...."
(L-179): "(ii)" should be "(iii)".
Thank you! We have corrected these oversights in the manuscript.

(Section 3.1): Parameters Γ1, Γ2, and A are shown in Fig. 3 and discussed in detail. But there is no description how the shape parameters λ1 and λ2 of the inverse-Gaussian distribution were as a result. It would be better to give us information about widths (Δ) or values of γ(=Δ2/Γ) of age spectra obtained by this method.

We thank the reviewer for pointing this out. The shape of the age spectrum is not well constrained by our method. This lack of constraint on the width of the age spectrum is now discussed in more detail in the text (see modifications to lines 236-254) and Figure S3 (see updated SI) was added to shows the pattern of all parameters fit by the algorithm.

From manuscript: "Unimodal age spectra are generally preferentially selected by the algorithm, in particular for the SH for young AoA (Fig. 3, panels (a)-(c)) but unimodal age spectra dominate the solution ensemble by a small margin except for AoA <0.5 years (Fig. S3). whereas bimodal spectra are slightly more common for older samples representing 50-80% of the solution ensemble in these bins. However, cConfidence intervals on age spectra parameters from bimodal spectra are considerably wider than for unimodal spectra. This implies that the parameters in a bimodal distribution are redundant and the shape of the age spectrum is not sufficiently constrained by the observations used. The width of the spectrum in each N$_2$O bin varies widely within the prescribed bounds for most unimodal and bimodal age spectra (Fig. S3). It appears that not enough data is available from the airborne campaigns to determine the amplitude of the seasonal cycle with enough confidence to constrain the width of the age spectrum and distinguish the relative contribution of the old peak (influencing only primarily the mean concentration difference to the troposphere) and the young peak (controlling the amplitude of seasonality) in bimodal age spectra. Because bimodal distributions are generated with a random value of the weighting factor A and the mean AoA of the second peak is assumed to be old (5-7 years), randomly generated bimodal solutions often produce overall AoA that is quite old. Therefore, they

~~are less likely to be selected by the MCMC algorithm for young AoA bins and a prevalent occurrence of unimodal age spectra in these bins is expected by chance. If more stratospheric data were available, the seasonality of CO₂ in each N₂O bin would be better resolved and the algorithm could derive tighter constraints on all parameters in bimodal age spectra, also allowing it to distinguish more clearly between unimodal or bimodal age spectra.~~ Additional observations with less scatter or information from different age tracers are needed to properly resolve the shape of the age spectra. Despite the limited resolution of the seasonal cycle, the observations are sufficient to place tight limits on the mean AoA in each N₂O bin and yield a well-characterised relationship between N₂O and mean AoA for each hemisphere."

(L-183 to Eq. 5): "....atmosphere δ$_{GST}$ with a molecular mass 1 amu greater than that of air ..." GST seems to be a kind of virtual tracer. It may be better to mention how the molecular diffusion volume was defined for D$_{GST}$.

The molecular diffusion volume is important in estimating the molecular diffusivity of a gas according to eq. B5. We chose D$_{GST}$ to be the diffusivity of Ar in air but the exact value of D$_{GST}$ has no influence on our results because results are subsequently normalized by D$_{GST}$ in equation 5. We clarified the phrasing of the manuscript (lines 212-218).

From manuscript: "To simplify the numerical treatment, we  simulate an idealized reference  tracer of gravitational fractionation  $\delta_{GST}$  which has a molecular mass 1 amu greater than  air and diffusivity equal that of Ar (see Appendix B & C). This tracer can be scaled offline to obtain the gravitational separation signal in any other species, including δ(Ar/N₂), e.g.

$$\delta(\text{Ar/N}_2) \approx \frac{(M_{Ar} - M_{air}) \times D_{Ar} - (M_{N2} - M_{air}) \times D_{N2}}{(M_{GST} - M_{air}) \times D_{GST}} \times \delta_{GST}.$$

The appropriate diffusivity values $D_{Ar}$ and $D_{N2}$ for Ar and N₂ in air are derived in Appendix B for a ternary mixture of Ar, O₂, and N₂, extending previous work (Ishidoya et al., 2013; Belikov et al., 2019). $\delta_{GST}$ is divided by $D_{GST}$ in eq. 5 and therefore does not depend on the exact value chosen for $D_{GST}$."

(L-236 to 240): Again, with this AoA-GS plots alone, it is somewhat difficult to understand the difference between NH and SH. Vertical distributions of AoA and GS may be helpful.

Please see our response regarding model-data comparisons and data quality differences above.

(L-276 and Fig. S2): "...the small seasonal cycle amplitude of δ(Ar/N2) <6 per meg..." Certainly the average seasonal amplitude seems to be small, but we can see large scatters. Fluctuations of Ar/N2 ratio in short time will be partly smoothed by the mixing process during the upward transport from the tropical upper troposphere to the lower stratosphere via TTL. Thus, it may not be a big obstacle to AoA estimation from Ar/N2. Problem is, rather, that this scatter is real atmospheric signal or not.

We thank the reviewer for drawing our attention to this. The large scatter is present only in the HIPPO campaigns shown in Figure S2, and not in ATom 2-4, illustrating recent advances in sampling. We have updated the figure to better illustrate the campaign-dependence of precision. We have also revised the manuscript to discuss limitations in data quality of earlier campaigns (see response to comments above) and updated Fig 4 to represent these campaign differences.

(Eq. 7 in Appendix A): This number should be something like "(A1)". Also please check eq. numbers in Appendix B and C.

Thank you! We updated the equation and table numbering and referencing in all appendices.

(L-411):"ΔMi the molecular mass difference to air "ΔMi appears in eq. (A21) for the first time.

We move the definition of $\Delta M_i$ to match its occurrence earlier in the text (lines 449-452).

From the manuscript: "Neglecting these smaller terms yields the governing Eq. (4) used in our model simulation

$$f_i \approx -ND_i \left[ \left( \frac{1}{H_i} - \frac{1}{H_{air}} \right) C_i \right] = -ND_i \Delta M_i \frac{g}{RT} C_i \qquad (B1)$$

where we introduced $\Delta M_i \equiv M_i - M_{air}$ the molecular mass difference to air (kg mol$^{-1}$) as a short-hand. Equation (BA21) is equally valid for trace gases such as Ar and major gases $N_2$ and $O_2$ when the appropriate diffusivities given by Eqs. (BA12) and (BA14) are used."

(Eqs. B1, B2, and B3): Please check the signs (plus/minus) of the molecular diffusion terms in these equations. In the conservation equation, term of the flux divergence (3rd term RHS in eq. B1) should be a form like -∇·[f] if we put it on RHS, just the same as the eddy diffusion term. The 3rd and 4th terms of RHS in eq. B2 might have the same signs after using the chain rule. The 3rd term of RHS in eq. B3 might have the same sign with 4th term of RHS in eq. B2 after eliminating a small term.

We thank the reviewer for catching this typo. Indeed, the gravitational flux divergence term should be positive and of the same sign as the eddy diffusion term.

(L-418): "D" in the expression of Peclet number should be "De".

Yes, the manuscript was corrected.

(L-432): "Eq. (6)" should be Eq. (5).

Yes, the manuscript was corrected.

**Comments from Reviewer 2**

This paper uses measured and simulated ratios of Ar/N2 to deduce gravitational separation in the stratosphere and their relationship to the stratospheric age of air. The Ar/N2 relationships to AoA in this study are relatively compact compared to the previous Belikov et al. study on gravitational separation and the agreement between the observations and model relationships are improved. This suggests that gravitational separation of Ar/N2, and also potentially the ratios of heavier noble gases, could be useful as an additional diagnostic of transport. The measurements, model runs and methodology are well described and the improvement in the treatment of gravitational separation in both the measurements and model compared to previous studies is an important step forward.

My main concern with this study is in the calculation of age of air from the measurements. The technique used, although rigorous, is based on assumptions that are not consistent with our current understanding

of the shape of age spectra and the transport of air into the lowermost stratosphere. The technique and assumptions are based on a study that is nearly two decades old and there have been numerous studies since that time refining our understanding of age spectra and how they can be inferred from measurements. The results may not change substantially by using more realistic age spectra and transport assumptions but that isn't entirely clear. Regardless, the more recent studies on this topic and the newer techniques should be acknowledged and shown to be in agreement with, or ideally replace, the results from the technique used in this study. Specific references and more detail on this topic are included below.

The results of this study are significant and relevant to ACP so I recommend publication after modification of the age of air calculation and consideration of the specific comments below.

We thank reviewer 2 for a favorable review and insightful comments on our manuscript. We made changes in how we present and treat the calculation of AoA and will discuss these in more detail in response to the specific comments below.

**Specific comments**

Line 35: should add 'however' before 'observational'

Thank you, we changed the manuscript accordingly.

Lines 40-41: should add references here to studies that have used tracers to infer age spectra (Andrews et al., JGR, 1999; Schoeberl et al., JGR, 2005; Hauck et al., ACP, 2019; Podglajen and Ploeger, ACP, 2019).

Thank you for the suggested references. We added them to the manuscript (lines 168-173).

From manuscript:"Though more complex representations of the age spectrum have recently been proposed, accounting for multiple modes and seasonal variations (e.g., Schoeberl et al., 2005; Li et al., 2012; Hauck et al., 2019, 2020; Podglajen and Ploeger, 2019), these require additional information from multiple tracers or models, and good seasonal data coverage that is not available here. In any case, we only rely on the first moment of the age distribution (i.e., the mean age) for calibrating the $N_2O$-AoA relationship, and the mean age is not particularly sensitive to the assumed shape of the age spectrum (Andrews et al., 2001)."

Line 94: did you mean 'CO2' instead of 'CO' or should 'CO2' be added here?

We used CO2 and CO from the Picarro and QCLS. The list was corrected (line 108).

Line 97: 'sample flask i to sample flask i+1'

Thank you, we changed the manuscript accordingly.

Line 108: change 'as a combined result of' to 'due to'

Thank you, we changed the manuscript accordingly.

Line 113: 'used' instead of 'use'

Thank you, we changed the manuscript accordingly.

Section 2.2: This section is my main concern with the paper as I mentioned above. I have two issues: (1) the assumed shape of the age spectra and (2) the tropical single entry point assumption for the lowermost stratosphere. The first issue, the shape of stratospheric age spectra has been shown in a number of studies derived from various model and reanalysis output based on trajectories or boundary impulse functions (Reithmeier et al., Clim. Dyn., 2008; Li et al., JGR, 2012; Diallo et al., ACP, 2012; Ray et al., JGR, 2014; Ploeger and Birner, ACP, 2016; Podglajen and Ploeger, ACP, 2019; Hauck et al., ACP, 2019). In these studies, it is clear that the dominant mode of variability in the age spectra in the lower stratosphere is the seasonal cycle, and for those models with a QBO, that is the second largest mode of variability. None of these results show a peak in the 5-7 year range relative to the 2-5 year range of the spectra. The use of two peak age spectra with one of the peaks in the 5-7 year range is not realistic for any part of the stratosphere and especially not in the lower stratosphere where the seasonal cycle is so large and the 5-7 year portion of the age spectra has much smaller values (often by an order of magnitude) compared to those at ages less than 5 years. The lack of reference in the manuscript to any of the papers listed above and the use of an age spectra shape that is not consistent with any of them is a clear deficiency. The use of a single trace gas, such as CO2, can only reveal a limited amount of the age spectra. It has been shown by Hauck et al. (2019) that at least 5-10 tracers with a range of lifetimes are necessary to resolve the important features and the true shape of the age spectra. The Andrews et al. (1999, 2001) studies were important steps at the time in our understanding of how trace gas observations could be used to infer features of the age spectra. But now that we have a detailed understanding of age spectra from models and reanalysis it isn't appropriate to use age spectra defined only by the fit to CO2. The second issue is related to the transport of air into the lowermost stratosphere (LMS) where most of the measurements in this study were taken. Many studies have shown that the LMS consists of a seasonally varying mixture of tropospheric and stratospheric air with some portion of the tropospheric air originating from the extratropics or tropical air that has bypassed the tropical tropopause (e.g. Ray et al., JGR, 1999; Hoor et al., ACP, 2004; Olsen et al., JGR, 2004; Boenisch et al., ACP, 2009; Skerlak et al., ACP, 2014). This precludes the use of a single-entry tropical tropopause time series to derive age of air in the LMS. I highly recommend a technique considering the above points be used here. Even if the resulting mean age values do not differ significantly, the technique would be consistent with our current understanding of transport in the lower stratosphere.

We thank Reviewer 2 for raising this important concern.

Our selection criteria for "stratospheric" samples ensures that we focus on air that is of stratospheric origin and has experienced limited mixing with the troposphere. Therefore, we believe the assumption of a single entry-point to the stratosphere when calculating AoA is imperfect but justified. Other methods (e.g., Hauck et al, 2020), rely on model results to constrain the contribution of different tropospheric source regions to air found in the lowermost stratosphere. However, these model estimates will be representative of air *typically* found in a given region, not necessarily of air in small-scale features that we specifically target here for its stratospheric signature. This is evident also in a comparison of observed AoA and AoA simulated by TOMCAT. For the oldest samples, observed AoA is typically higher than simulated AoA (Fig. S4). This key feature of our sample selection is discussed in more detail in the manuscript now (lines 141-148) and we have lowered the $H_2O$ threshold value to 15 ppm in order to more carefully select for stratospheric composition. We now also emphasize in the text that our sampling strategy does not yield air that is fully representative of the lowermost stratosphere (lines 146-147 & 266-269).

Furthermore, the fact that our samples are typically older means that diabatic dispersion (Sparling et al., doi.org/10.1029/97JD01968, 1997) has acted on the age distribution and likely smoothed out the multiple peaks found in model studies.

Only a limited number of tracers are available across all campaigns and incorporating multiple new tracers into the method is beyond the scope of this manuscript. Therefore, constraining the full age spectrum is impossible as discussed by Reviewer 2. We adjusted the language throughout the manuscript to reflect that our method only really constrains the mean AoA of a sample and included additional references in the manuscript to studies that illustrate a more thorough treatment of the shape of the age spectrum. We also adjusted the range of possible mean AoA values of the second peak to range from 1-6 years. This and changing the $H_2O$ threshold does not substantially change our results.

Most importantly, our objective - establishing a relationship between $\delta(Ar/N_2)$ and mean AoA – is currently not limited by our ability to determine the AoA for these samples. AoA from $N_2O$ is still relatively more precise than our ability to sample and analyze air for $\delta(Ar/N_2)$. Moreover, our recalibration of the $N_2O$-AoA relationship which underlies our AoA calculations for Medusa samples is not overly sensitive to small errors introduced at young AoA where the risk of a strong tropospheric influence is greatest because the $N_2O$-AoA relationship is anchored at an age of zero to the mean concentration $N_2O$.

[revised manuscript text omitted]

Line 152: Gamma1 is repeated, the second one should be Gamma2.

Thank you, we corrected the typo.

Line 179: the second '(ii)' should be '(iii)'

Thank you, we corrected the typo.

Line 203: 'relationships'

Thank you, we corrected the typo.

Line 262: 'descends' instead of 'is sinking'

We adopted the suggested language.

Line 328: Is the Ar/N2 ratio 'mostly unaffected by seasonality' throughout the stratosphere?

This statement refers to the tropical upper troposphere. We clarified the phrasing to reflect this (line 371).

Lines 330-331: It's not so much that changes in AoA in the upper stratosphere and mesosphere are difficult to resolve from transient tracers but more that AoA becomes nearly constant in this region as you show in Figure 6. So AoA loses sensitivity in this region where gravitational separation gains sensitivity. This is a difficult region to make measurements of any kind so it may also be challenging to obtain enough measurements of noble gases to monitor gravitational separation.

Thank you. We adjusted the phrasing to reflect this distinction more clearly (lines 373-375). Indeed, sampling of the mesosphere is very challenging but we are hopeful that more careful noble gas

observations will become available in the next years from balloon samplers that can reach up to 30-35 km providing additional constraints on circulation in the lower and middle stratosphere.

[revised manuscript text omitted]

$$P(d|k,m) = \frac{1}{\left[(2\pi)^n \det\left(\widehat{C}_e\right)\right]^{0.5}} \exp\left(-\frac{1}{2} e^T \widehat{C}_e^{-1} e\right) \qquad \text{(A1)}$$

where $\widehat{C}_e$ is the covariance matrix.  We assume that all $n$ observations are independent. Thus, $\widehat{C}_e$ has only diagonal entries of $\sigma_{CO_2}^2$ and $\det\left(\widehat{C}_e\right)$ simplifies to $\left(\sigma_{CO_2}^2\right)^n$. The value of $\sigma_{CO_2}^2$ is different for each bin and determined iteratively as the approximate root mean square error of the observations around the final time series for each bin obtained at the end of the MCMC algorithm. Typical values of $\sigma_{CO_2}^2$ are between 0.18 and 1.28 ppm and generally decrease with increasing AoA of a $N_2O$ bin.

6) If this is the first pass of the chain, define $k_{saved}$ and $m_{saved}$ to equal $k$ and $m$. Otherwise, calculate the selection criterion $\alpha \equiv \min\left(1, \frac{P(d|k,m)}{P(d|k_{saved},m_{saved})}\right)$ and accept $k$ and $m$ as new saved values ($k_{saved}, m_{saved}$) with probability $\alpha$. Sampling from the same prior distributions on each pass of the chain simplifies our expression of $\alpha$ compared to that presented by Malinverno (2002), making it only dependent on the likelihood ratio.

7) Repeat steps 2-6 1000 times sampling parameter values from the same prior distributions and store the final values of $k_{saved}$ and $m_{saved}$ obtained after the 1000[th] iteration (i.e., a plausible solution produced past the burn-in period) for later use.

8) To sample the full posterior pdf (i.e., the full uncertainty about the age spectrum parameters), initialize 2000 different Markov chains by repeating steps 1-7. Each stored value of $\boldsymbol{m}$ characterizes one age spectrum that is likely not far from the best solution, given the data $\boldsymbol{d}$, yielding an ensemble of 2000 age spectra from which statistics can be computed. Note that each Markov chain is fully independent, so the algorithm can be easily parallelized to minimize computational costs.

**8. Appendix B: Derivation of Eq. (4) from the Maxwell-Stefan Equations**

We start by approximating air as a ternary mixture of $N_2$, $O_2$ and Ar, and later generalize to consider additional trace species. According the Maxwell-Stefan equations (Taylor and Krishna, 1993) diffusion in this ternary mixture is governed by:

$$d_{N2} = \frac{C_{N2}\, f_{Ar} - C_{Ar}\, f_{N2}}{N \times D_{N2:Ar}} + \frac{C_{N2}\, f_{O2} - C_{O2}\, f_{N2}}{N \times D_{N2:O2}} \qquad \text{(B1)}$$

$$d_{Ar} = \frac{C_{Ar}\, f_{N2} - C_{N2}\, f_{Ar}}{N \times D_{Ar:N2}} + \frac{C_{Ar}\, f_{O2} - C_{O2}\, f_{Ar}}{N \times D_{Ar:O2}} \qquad \text{(B2)}$$

$$f_{O2} = -f_{N2} - f_{Ar} \qquad \text{(B3)}$$

[revised manuscript text omitted]

**δ(Ar/N₂) residual analysis**

To test the hypothesis that large artifacts or biases in δ(Ar/N₂) were introduced during flask sample collection, we look for the presence of a pressure dependency of δ(Ar/N₂) after removing what we have shown is the natural gravitational signal. Pressure serves as a proxy for sampling conditions and aircraft velocity simultaneously here because aircraft velocity also increases systematically with altitude or pressure altitude. A vertical profile of δ(Ar/N₂) (Fig. S1A) reveals no strong dependency on pressure in the troposphere below about 300 mb. Above ~300 mb, some samples represent stratospheric conditions and therefore have an age of air (AoA) greater than zero and more negative δ(Ar/N₂). For further analysis, we correct δ(Ar/N₂) for the natural gravitational signal ($\delta(Ar/N_2)_{grav}$) using observed AoA from each sample and a quadratic fit of the AoA-δ(Ar/N₂) relationship simulated in TOMCAT ($\delta(Ar/N_2)_{grav} = -8.124\,AoA^2 - 22.73\,AoA + 0.9529$). The result is shown in Figure S1B. Although there is scatter in the corrected data, in particular for HIPPO1−5, no systematic relationship between pressure and gravitationally-corrected δ(Ar/N₂) is evident, supporting the interpretation that aircraft sampling did not create large biases in δ(Ar/N₂). While there is some evidence of a pressure dependence in the ORCAS data from the Southern Hemisphere (SH), this could also be the result of an incomplete correction for gravitational fractionation if there are differences in the true AoA-δ(Ar/N₂) relationship for the SH not captured by TOMCAT. However, stratospheric samples from ORCAS were almost exclusively taken at the same pressure, so our analysis of the δ(Ar/N₂)-AoA relationship should be robust regardless.

[Figure]

**Figure S1. Pressure relationship of Medusa flask δ(Ar/N₂) (A) and residual δ(Ar/N₂) (B), i.e., δ(Ar/N₂) corrected for the influence of gravity in the stratosphere (see text). Data from HIPPO (open circles) and ATom (solid circles), and ORCAS (solid diamonds) are shown separately in (B) with their respective error bars.**

[Figure]

[Figure]

**Figure S2. Seasonal cycle of δ(Ar/N₂) in the topical  free tropospheric (20°>lat>-20° & altitude 3-8 km).**

[Figure]

45 **Figure S3. Age spectrum parameter values selected by the Markov chain Monte Carlo algorithm in each N$_2$O bin for the Southern (blue) and Northern (red) Hemispheres with error bars showing 95% confidence interval. NH N$_2$O bin values were adjusted by -1.5 ppb here for visual clarity only. The mean AoA is well constrained but the algorithm fails to clearly distinguish between unimodal and bimodal age spectra and places poor constraints on the width of unimodal and bimodal age spectra.**

[Figure]

**Figure S4**. Comparison of  δ(Ar/N₂) (left) and AoA (right) between stratospheric HIPPO, AToM and ORCAS observations and TOMCAT  in the lowermost stratosphere. TOMCAT grid cells are chosen to be closest in space in time to the observations, but unresolved mixing processes  can allow more fractionated/older air to be present in the observations than can be reproduced by TOMCAT. These deviations from the 1:1-line are correlated between δ(Ar/N₂) and AoA, and therefore will partially cancel in the AoA-δ(Ar/N₂)  relationship (see Fig. 5 in main paper).

[Figure]

**Figure S5. Width of the 95% confidence interval for AoA estimates from $N_2O$ based on the Markov chain Monte Carlo algorithm (points) and from $\delta(Ar/N_2)$ using the AoA-$\delta(Ar/N_2)$ relationship in TOMCAT (red line).**

60